# Gradient-Variation Online Adaptivity for Accelerated Optimization with Hölder Smoothness

**Yuheng Zhao**[1,2], **Yu-Hu Yan**[1,2], **Kfir Yehuda Levy**[3], **Peng Zhao**[1,2]

[1] National Key Laboratory for Novel Software Technology, Nanjing University, China
[2] School of Artificial Intelligence, Nanjing University, China
[3] Electrical and Computer Engineering, Technion, Haifa, Israel

## Abstract

Smoothness is known to be crucial for acceleration in offline optimization, and for gradient-variation regret minimization in online learning. Interestingly, these two problems are actually closely connected — accelerated optimization can be understood through the lens of gradient-variation online learning. In this paper, we investigate online learning with *Hölder smooth* functions, a general class encompassing both smooth and non-smooth (Lipschitz) functions, and explore its implications for offline optimization. For (strongly) convex online functions, we design the corresponding gradient-variation online learning algorithm whose regret smoothly interpolates between the optimal guarantees in smooth and non-smooth regimes. Notably, our algorithms do not require prior knowledge of the Hölder smoothness parameter, exhibiting strong adaptivity over existing methods. Through online-to-batch conversion, this gradient-variation online adaptivity yields an optimal universal method for stochastic convex optimization under Hölder smoothness. However, achieving universality in offline strongly convex optimization is more challenging. We address this by integrating online adaptivity with a detection-based guess-and-check procedure, which, for the first time, yields a universal offline method that achieves accelerated convergence in the smooth regime while maintaining near-optimal convergence in the non-smooth one.

## 1 Introduction

First-order optimization methods based on (stochastic) gradients are widely used in machine learning due to their efficiency and simplicity [Nesterov, 2018; Duchi et al., 2011; Kingma and Ba, 2015]. It is well-known that the curvature of the objective function strongly influences the difficulty of optimization. In particular, the optimal convergence rates differ significantly between smooth and non-smooth objectives. For convex functions, the optimal rate in the non-smooth case is $\mathcal{O}(1/\sqrt{T})$, achievable by standard gradient descent (GD), where $T$ denotes the total number of gradient queries. In contrast, for smooth functions, GD only attains an $\mathcal{O}(1/T)$ rate, which exhibits a large gap with the accelerated rate $\mathcal{O}(1/T^2)$ attained by the Nesterov's accelerated gradient (NAG) method [Nesterov, 2018]. Similar acceleration phenomena also arise in the strongly convex setting.

The significant performance gap between smooth and non-smooth optimization has motivated the study of *universality* in optimization [Nesterov, 2015]: an ideal universal method should adapt to both unknown smooth and non-smooth cases, achieving optimal convergence in both regimes. Several studies have explored adapting to a more challenging setting known as *Hölder smoothness* [Devolder et al., 2014; Nesterov, 2015], which continuously interpolates between smooth and non-smooth

---

*Correspondence: Peng Zhao <zhaop@lamda.nju.edu.cn>

Table 1: Summary and comparison of the convergence rates of existing universal methods and ours for offline (strongly) convex optimization. "Weak/strong universality" notations are defined in Definition 1. Let $\kappa \triangleq L/\lambda$. $\sigma$ denotes the variance in the stochastic setting. The rate marked with † is non-accelerated in smooth functions.

| Setting | Convergence Rate | Universality |
|---|---|---|
| **Convex** | $\mathcal{O}(\frac{L}{T^2} + \frac{\sigma}{\sqrt{T}})$ for $L$-smooth; $\mathcal{O}(\frac{1}{\sqrt{T}})$ for Lipschitz [Kavis et al., 2019] | Weak |
| | $\mathcal{O}\left(\frac{L_\nu}{T^{(1+3\nu)/2}} + \frac{\sigma}{\sqrt{T}}\right)$ for $(L_\nu, \nu)$-Hölder smooth [Rodomanov et al., 2024] | Strong |
| | $\mathcal{O}\left(\frac{L_\nu}{T^{(1+3\nu)/2}} + \frac{\sigma}{\sqrt{T}}\right)$ for $(L_\nu, \nu)$-Hölder smooth [Theorem 2] | Strong |
| $\lambda$-**Strongly Convex** | $\mathcal{O}\left(\exp(-\frac{T}{\kappa}) \cdot \frac{T}{\kappa}\right)$ for $L$-smooth and Lipschitz; $\widetilde{\mathcal{O}}(\frac{1}{\lambda T})$ for Lipschitz [Levy, 2017] | Weak† |
| | $\mathcal{O}\left(\exp(-\frac{T}{6\sqrt{\kappa}})\right)$ for $L$-smooth and Lipschitz; $\widetilde{\mathcal{O}}(\frac{1}{\lambda T})$ for Lipschitz [Theorem 4] | Weak |

functions. Formally, a function $\ell : \mathbb{R}^d \to \mathbb{R}$ is $(L_\nu, \nu)$-Hölder smooth with respect to the $\ell_2$-norm, where $L_\nu > 0$ and $\nu \in [0, 1]$, if

$$\|\nabla \ell(\mathbf{x}) - \nabla \ell(\mathbf{y})\|_2 \leq L_\nu \|\mathbf{x} - \mathbf{y}\|_2^\nu, \quad \forall \mathbf{x}, \mathbf{y} \in \mathbb{R}^d. \tag{1}$$

It can be observed that $L_\ell$-smoothness corresponds to $(L_\ell, 1)$-Hölder smoothness, while $G$-Lipschitz continuity is $(2G, 0)$-Hölder smoothness. For convex and $(L_\nu, \nu)$-Hölder smooth functions, the optimal convergence rate is $\mathcal{O}(1/T^{(1+3\nu)/2})$ and has been achieved by several recent methods [Nesterov, 2015; Li and Lan, 2025; Rodomanov et al., 2024]. However, progress in the strongly convex setting remains limited, even adapting only between the smooth and non-smooth cases is still unresolved. Levy [2017] attained a non-accelerated rate in the smooth case and a suboptimal rate in the non-smooth case. It remains open on how to achieve universality in the strongly convex setting, particularly in obtaining an accelerated rate in the smooth regime.

**Online-to-Batch Conversion.** An important perspective for designing optimization algorithms is the *Online-to-Batch Conversion* framework [Cesa-Bianchi et al., 2004], which reformulates an offline optimization problem as a regret minimization task addressed by online learning algorithms. The key advantage typically lies in the simplicity of the converted algorithm. More importantly, it allows one to leverage the rich adaptivity results developed in online learning to enhance optimization performance. In online learning, the goal is to minimize the *regret* over a sequence of $T$ online functions, denoted by $\text{REG}_T$ [Hazan, 2016]. For convex functions, the standard Online-to-Batch (O2B) conversion implies that the convergence rate of the averaged iterate is $\text{REG}_T/T$. Consequently, in the non-smooth case, combining this conversion with an online algorithm achieving an optimal $\mathcal{O}(\sqrt{T})$ regret directly yields the optimal convergence rate for offline optimization of order $\mathcal{O}(1/\sqrt{T})$.

In offline smooth optimization, achieving optimal convergence requires more refined conversion techniques and adaptive online algorithms [Levy, 2017; Cutkosky, 2019]. Specifically, attaining the optimal $\mathcal{O}(1/T^2)$ rate for smooth functions relies on an O2B conversion with *stabilized gradient evaluations* [Cutkosky, 2019] and an online learning algorithm equipped with *gradient-variation adaptivity* [Chiang et al., 2012; Zhao et al., 2020]. Notably, Kavis et al. [2019] achieved optimal rates in both smooth and non-smooth cases without requiring prior knowledge of smoothness, thereby attaining universality. A concise technical discussion can be found in the lecture note [Zhao, 2025].

However, a gap persists in online-to-offline methods under general Hölder smoothness settings. Furthermore, for strongly convex functions, achieving universality remains far from complete, even when focusing solely on smooth and non-smooth cases. These challenges motivate our study of *gradient-variation online learning with Hölder smoothness* and the development of effective conversion techniques to offline optimization.

**Our Contributions.** Our contributions are mainly two-fold, summarized as follows.

*(i)* For convex functions, we study gradient-variation online learning with Hölder smoothness, achieving an $\mathcal{O}(\sqrt{V_T} + L_\nu T^{\frac{1-\nu}{2}})$ regret that seamlessly interpolates between the optimal guarantees in smooth and non-smooth regimes. Leveraging this adaptivity through O2B conversion, we achieve universality for stochastic convex optimization with Hölder smoothness, matching the optimal result [Rodomanov et al., 2024] obtained using arguably more sophisticated techniques.

*(ii)* For strongly convex functions, we develop an $\mathcal{O}\big(\frac{1}{\lambda} \log V_T + \frac{1}{\lambda} L_\nu^2 (\log T)^{\frac{1-\nu}{1+\nu}}\big)$ gradient-variation regret with Hölder smoothness that recovers the optimal results in both smooth and non-smooth scenarios. Achieving universality in offline strongly convex optimization presents additional challenges. We address this by integrating online adaptivity with a detection-based guess-and-check procedure. Combined with a carefully designed O2B conversion, for the first time, we provide a universal method for strongly convex optimization that achieves *accelerated* convergence in the smooth regime while maintaining near-optimal convergence in the non-smooth one.

The convergence rates of the existing universal methods and our newly obtained results are summarized in Table 1. Our work opens a new avenue for converting gradient-variation online adaptivity to offline optimization, and recent progress in gradient-variation online learning [Zhao et al., 2024; Yan et al., 2024] suggests possible further opportunities. Moreover, we hope it will inspire broader efforts to integrate online adaptivity into offline methods, which may not only advance the pursuit of universality and other forms of adaptivity in offline optimization but also shed light on the design and understanding of modern optimizers in deep learning [Cutkosky et al., 2023; Chen and Hazan, 2024].

**Organization.** The rest is organized as follows. Section 2 introduces the problem setup and preliminaries. Section 3 presents our results for convex functions in both online and offline settings. Section 4 considers the strongly convex scenario, where achieving universality in offline optimization is particularly challenging. Section 5 concludes the paper. All proofs are in the appendices.

## 2 Problem Setups and Preliminaries

This section provides preliminaries. Section 2.1 introduces the problem setup of offline (stochastic) first-order optimization. Section 2.2 covers online learning and its gradient-variation adaptivity. Section 2.3 introduces online-to-batch conversion and its advanced variants.

**Notations.** We use $[N]$ to denote the index set $\{1, 2, \ldots, N\}$. The shorthand $\sum_t$ stands for $\sum_{t \in [T]}$, and we define $\sum_{i=a}^b c_i = 0$ whenever $a > b$. The Bregman divergence associated with a convex regularizer $\psi : \mathcal{X} \to \mathbb{R}$ is defined as $\mathcal{D}_\psi(\mathbf{x}, \mathbf{y}) \triangleq \psi(\mathbf{x}) - \psi(\mathbf{y}) - \langle \nabla \psi(\mathbf{y}), \mathbf{x} - \mathbf{y} \rangle$. By default, $\|\cdot\|$ denotes the $\ell_2$-norm. We use $a \lesssim b$ to indicate $a = \mathcal{O}(b)$, and use $\widetilde{\mathcal{O}}(\cdot)$-notation to suppress poly-logarithmic factors in the leading terms.

### 2.1 Offline Optimization: Acceleration and Universality

Consider the optimization problem over a convex feasible domain $\mathcal{X} \subseteq \mathbb{R}^d$,

$$\min_{\mathbf{x} \in \mathcal{X}} \ell(\mathbf{x}), \tag{2}$$

where $\ell : \mathcal{X} \to \mathbb{R}$ is a convex objective. We assume the algorithm has access to a *gradient oracle* denoted by $\mathbf{g}(\cdot)$, and consider two settings:

*(i)* *Deterministic* setting: $\mathbf{g}(\cdot)$ returns the exact gradient, i.e., $\mathbf{g}(\mathbf{x}) = \nabla \ell(\mathbf{x})$.
*(ii)* *Stochastic* setting: $\mathbf{g}(\cdot)$ provides an unbiased estimate of the gradient, $\mathbb{E}[\mathbf{g}(\mathbf{x}) \mid \mathbf{x}] = \nabla \ell(\mathbf{x})$, and satisfies the standard bounded-variance condition, $\mathbb{E}[\|\mathbf{g}(\mathbf{x}) - \nabla \ell(\mathbf{x})\|^2 \mid \mathbf{x}] \leq \sigma^2, \forall \mathbf{x} \in \mathcal{X}$.

Suppose the algorithm is allowed $T$ queries to the gradient oracle and outputs a final solution $\mathbf{x}_T^\dagger$. We focus on the convergence rates of the sub-optimality gap, i.e., $\ell(\mathbf{x}_T^\dagger) - \min_{\mathbf{x} \in \mathcal{X}} \ell(\mathbf{x}) \leq \varepsilon_T$.

It is well known that smoothness plays a central role in *accelerated convergence* [Nesterov, 2018]. Consider the deterministic setting as an example. For convex functions, the optimal convergence rate is $\mathcal{O}(1/\sqrt{T})$ for Lipschitz objectives, which can be accelerated to $\mathcal{O}(1/T^2)$ when the objective is smooth. For strongly convex functions, the optimal rate improves from $\mathcal{O}(1/T)$ in the Lipschitz case to $\mathcal{O}(\exp(-T/\sqrt{\kappa}))$ for smooth objectives, where $\kappa \triangleq L_\ell/\lambda$ denotes the condition number.

Prior research has aimed to develop a single algorithm that can adaptively achieve (optimal) guarantees without prior knowledge of whether the objective is smooth or non-smooth. In addition, several studies have extended this adaptivity to the broader setting of Hölder smoothness. This adaptability, known as *universality* in optimization methods [Nesterov, 2015], has attracted considerable attention in recent years [Levy, 2017; Kavis et al., 2019; Rodomanov et al., 2024; Kreisler et al., 2024].

In this paper, to clearly distinguish the degrees of adaptability of optimization methods to different smoothness levels, we introduce the following definitions of *weak/strong universality*.

**Definition 1** (Weak/Strong Universality). An optimization method is said to be *universal* if it can automatically adapt to an *unknown* level of smoothness of the objective function. Specifically,

*(i)* **Weak universality:** it simultaneously adapts to smooth and non-smooth (Lipschitz) functions;

*(ii)* **Strong universality:** it simultaneously adapts to $(L_\nu, \nu)$-Hölder smooth functions for $\nu \in [0, 1]$.

It is infeasible to rely on the knowledge of smoothness parameter $L$ or Lipschitz continuity constant $G$ when developing weakly universal methods, and likewise on $\nu$ or $L_\nu$ when devising strongly universal methods. In essence, universality demands that the optimization method can automatically adapt to various scenarios, with weak universality adapting to two cases (smooth and Lipschitz) and strong universality extending to broader Hölder smoothness.

## 2.2 Online Optimization: Regret and Gradient-Variation Adaptivity

Online Convex Optimization (OCO) [Hazan, 2022] is a versatile online learning framework, typically modeled as an iterative game between a player and the environment. At iteration $t \in [T]$, the player chooses a decision $\mathbf{x}_t$ from a convex feasible domain $\mathcal{X} \subseteq \mathbb{R}^d$. Simultaneously, the environment reveals a convex function $f_t : \mathcal{X} \to \mathbb{R}$, and the player incurs a loss $f_t(\mathbf{x}_t)$. The player then receives the gradient information to update $\mathbf{x}_{t+1}$, aiming to optimize the *regret* defined as

$$\text{REG}_T \triangleq \sum_{t=1}^{T} f_t(\mathbf{x}_t) - \min_{\mathbf{x} \in \mathcal{X}} \sum_{t=1}^{T} f_t(\mathbf{x}). \tag{3}$$

For *Lipschitz* online functions, the minimax optimal regret bounds are $\mathcal{O}(\sqrt{T})$ for convex functions [Zinkevich, 2003] and $\mathcal{O}(\frac{1}{\lambda} \log T)$ for $\lambda$-strongly convex functions [Hazan et al., 2007]. When the functions are *smooth*, we can further obtain *problem-dependent* regret guarantees, which enjoy better bounds in easy problem instances while maintaining the same minimax optimality in the worst case [de Rooij et al., 2014; Foster et al., 2015]. Among many problem-dependent quantities, a particular one called *gradient variations* draws much attention [Chiang et al., 2012; Yang et al., 2014], which is defined to capture how the gradients of online functions evolve over time,

$$V_T \triangleq \sum_{t=2}^{T} \sup_{\mathbf{x} \in \mathcal{X}} \|\nabla f_t(\mathbf{x}) - \nabla f_{t-1}(\mathbf{x})\|^2. \tag{4}$$

It is established that optimal gradient-variation regret for convex and $\lambda$-strongly convex functions are $\mathcal{O}(\sqrt{V_T})$ and $\mathcal{O}(\frac{1}{\lambda} \log V_T)$ [Chiang et al., 2012], respectively. There has been significant subsequent development in more complex environments [Zhao et al., 2020, 2024; Sachs et al., 2023; Yan et al., 2023, 2024; Xie et al., 2024]. Gradient-variation online learning has gained significant attention due to its impact on analyzing trajectory dynamics and its fundamental connections to various optimization problems. It has been proved essential for fast convergence in minimax games [Syrgkanis et al., 2015; Zhang et al., 2022] and bridging adversarial and stochastic convex optimization [Sachs et al., 2022; Chen et al., 2024]. Recent results also demonstrate its important role in accelerated optimization [Cutkosky, 2019; Kavis et al., 2019; Joulani et al., 2020b].

## 2.3 Online-to-Batch Conversion: Stabilization

Consider the optimization problem of $\min_{\mathbf{x} \in \mathcal{X}} \ell(\mathbf{x})$ with access to a (stochastic) gradient oracle $\mathbf{g}(\cdot)$. This problem can be solved using online algorithms with online-to-batch conversion. A basic example is as follows: we define the online function $f_t(\mathbf{x}) \triangleq \langle \mathbf{g}(\mathbf{x}_t), \mathbf{x} \rangle$ and ensure that for any $\mathbf{x}_\star \in \mathcal{X}$:

$$\mathbb{E}\left[\ell\left(\frac{1}{T} \sum_{t=1}^{T} \mathbf{x}_t\right)\right] - \ell(\mathbf{x}_\star) \leq \frac{1}{T} \mathbb{E}\left[\sum_{t=1}^{T} \langle \mathbf{g}(\mathbf{x}_t), \mathbf{x}_t - \mathbf{x}_\star \rangle\right] \leq \frac{1}{T} \mathbb{E}\left[\text{REG}_T\right].$$

Hence, for convex objectives, if the online algorithm achieves a regret bound of $\mathcal{O}(\sqrt{T})$, the corresponding offline optimization method directly attains a convergence rate of $\mathcal{O}(1/\sqrt{T})$.

To achieve accelerated rates in smooth optimization, advanced conversion methods and adaptive online algorithms are required. The key insight is to evaluate the gradient on weighted averaged iterates, which introduces a *stabilization* effect [Wang and Abernethy, 2018; Cutkosky, 2019].

---

**Algorithm 1** Stabilized Online-to-Batch Conversion

---

**Input:** Online learning algorithm $\mathcal{A}_{\mathrm{OL}}$, weights $\{\alpha_t\}_{t=1}^T$ with $\alpha_t > 0$.
1: **Initialization:** $\mathbf{x}_1 \in \mathcal{X}$.
2: **for** $t = 1$ **to** $T$ **do**
3:     Calculate $\overline{\mathbf{x}}_t = \frac{1}{\alpha_{1:t}} \sum_{s=1}^t \alpha_s \mathbf{x}_s$ with $\alpha_{1:t} \triangleq \sum_{s=1}^t \alpha_s$, receive $\mathbf{g}(\overline{\mathbf{x}}_t)$
4:     Construct $f_t(\mathbf{x}) \triangleq \alpha_t \langle \mathbf{g}(\overline{\mathbf{x}}_t), \mathbf{x} \rangle$ to $\mathcal{A}_{\mathrm{OL}}$ as the $t$-th iteration online function
5:     Obtain $\mathbf{x}_{t+1}$ from $\mathcal{A}_{\mathrm{OL}}$
6: **end for**
**Output:** $\overline{\mathbf{x}}_T = \frac{1}{\alpha_{1:T}} \sum_{t=1}^T \alpha_t \mathbf{x}_t$

---

**Stabilized Online-to-Batch Conversion [Cutkosky, 2019].** Algorithm 1 summarizes the conversion. Given an online learning algorithm $\mathcal{A}_{\mathrm{OL}}$ and a sequence of positive weights $\{\alpha_t\}_{t=1}^T$, the conversion operates as follows: At each iteration $t$, it computes a *weighted average* of past decisions $\overline{\mathbf{x}}_t = \frac{1}{\alpha_{1:t}} \sum_{s=1}^t \alpha_s \mathbf{x}_s$ with $\alpha_{1:t} \triangleq \sum_{s=1}^t \alpha_s$. It then queries the gradient $\mathbf{g}(\overline{\mathbf{x}}_t)$, and constructs the online function $f_t(\mathbf{x}) \triangleq \alpha_t \langle \mathbf{g}(\overline{\mathbf{x}}_t), \mathbf{x} \rangle$. This function $f_t(\cdot)$ is passed to the online algorithm $\mathcal{A}_{\mathrm{OL}}$ to obtain the next decision $\mathbf{x}_{t+1}$. After $T$ iterations, the conversion outputs the final decision $\overline{\mathbf{x}}_T = \frac{1}{\alpha_{1:T}} \sum_{t=1}^T \alpha_t \mathbf{x}_t$. The conversion ensures the following inequality holds for all $\mathbf{x}_\star \in \mathcal{X}$:

$$\mathbb{E}\left[\ell(\overline{\mathbf{x}}_T)\right] - \ell(\mathbf{x}_\star) \leq \frac{1}{\alpha_{1:T}} \mathbb{E}\left[\sum_{t=1}^T \alpha_t \langle \mathbf{g}(\overline{\mathbf{x}}_t), \mathbf{x}_t - \mathbf{x}_\star \rangle\right] = \frac{\mathbb{E}\left[\mathrm{REG}_T^{\boldsymbol{\alpha}}\right]}{\alpha_{1:T}}, \tag{5}$$

where the expectation is taken over gradient randomness, and $\mathrm{REG}_T^{\boldsymbol{\alpha}} \triangleq \sum_{t=1}^T \alpha_t \langle \mathbf{g}(\overline{\mathbf{x}}_t), \mathbf{x}_t - \mathbf{x}_\star \rangle$ is the weighted regret of the online learning algorithm $\mathcal{A}_{\mathrm{OL}}$.

This conversion of Eq. (5) enables accelerated convergence for convex and smooth optimization. For example, by setting $\alpha_t = t$ and leveraging gradient-variation online adaptivity to obtain an $\mathcal{O}(1)$ weighted regret, Kavis et al. [2019] ultimately achieved a convergence rate of $\mathcal{O}(1/T^2)$.

## 3 Convex Optimization with Hölder Smoothness

We achieve the gradient-variation regret bound with Hölder smoothness in Section 3.1, then apply our method to obtain the universal method for stochastic convex optimization in Section 3.2.

### 3.1 Gradient-Variation Online Learning with Hölder Smoothness

We aim to establish gradient-variation regret for online learning with convex and $(L_\nu, \nu)$-Hölder smooth functions $\{f_t\}_{t=1}^T$. A commonly used bounded domain assumption is required [Hazan, 2022].

**Assumption 1** (Bounded Domain). The feasible domain $\mathcal{X} \subseteq \mathbb{R}^d$ is non-empty and closed with the diameter bounded by $D$, that is, $\|\mathbf{x} - \mathbf{y}\|_2 \leq D$ for all $\mathbf{x}, \mathbf{y} \in \mathcal{X}$.

We leverage the optimistic online gradient descent (optimistic OGD) [Chiang et al., 2012] as our algorithmic framework for gradient-variation online learning. Optimistic OGD is similar to online gradient descent [Zinkevich, 2003], e.g., $\widehat{\mathbf{x}}_{t+1} = \widehat{\mathbf{x}}_t - \eta \nabla f_t(\widehat{\mathbf{x}}_t)$, but a key difference lies in the point where the gradient $\nabla f_t(\cdot)$ is evaluated. In optimistic OGD, the gradient is computed at a point $\mathbf{x}_t$, which is updated one step ahead of $\widehat{\mathbf{x}}_t$ using a prediction $M_t \in \mathbb{R}^d$ for the upcoming gradient. To this end, optimistic OGD maintains two decision sequences $\{\mathbf{x}_t\}_{t=1}^T, \{\widehat{\mathbf{x}}_t\}_{t=1}^T$, and updates by

$$\mathbf{x}_t = \Pi_{\mathcal{X}}\left[\widehat{\mathbf{x}}_t - \eta_t M_t\right], \quad \widehat{\mathbf{x}}_{t+1} = \Pi_{\mathcal{X}}\left[\widehat{\mathbf{x}}_t - \eta_t \nabla f_t(\mathbf{x}_t)\right], \tag{6}$$

where $\eta_t > 0$ is a time-varying step size, and $\Pi_{\mathcal{X}}[\mathbf{y}] \triangleq \arg\min_{\mathbf{x} \in \mathcal{X}} \|\mathbf{x} - \mathbf{y}\|_2$ is Euclidean projection.

We first review the derivation of the $\mathcal{O}(\sqrt{V_T})$ gradient-variation bound under the $L$-smoothness assumption on $\{f_t\}_{t=1}^T$. By setting $M_t = \nabla f_{t-1}(\mathbf{x}_{t-1})$, optimistic OGD yields the following classical gradient-variation analysis [Chiang et al., 2012]:

$$\mathrm{REG}_T \lesssim \frac{1}{\eta_T} + \sum_{t=1}^T \eta_t \|\nabla f_t(\mathbf{x}_t) - \nabla f_{t-1}(\mathbf{x}_{t-1})\|^2 - \sum_{t=2}^T \frac{1}{\eta_{t-1}} \|\mathbf{x}_t - \mathbf{x}_{t-1}\|^2. \tag{7}$$

Given the smoothness parameter $L$, deriving $\mathcal{O}(\sqrt{V_T})$ from Eq. (7) is straightforward by appropriately setting the step size $\eta_t$. On the right-hand side, the second term $\sum_t \eta_t \|\nabla f_t(\mathbf{x}_t) - \nabla f_{t-1}(\mathbf{x}_{t-1})\|^2$ is an adaptivity term measuring the deviation between the two gradients, and the last one is a negative stability term. The adaptivity term can be bounded by $\|\nabla f_t(\mathbf{x}_t) - \nabla f_{t-1}(\mathbf{x}_t)\|^2 + \|\nabla f_{t-1}(\mathbf{x}_t) - \nabla f_{t-1}(\mathbf{x}_{t-1})\|^2$, where the first part can be converted to the desired gradient variation Eq. (4) and the second part is bounded by $L^2 \|\mathbf{x}_t - \mathbf{x}_{t-1}\|^2$ under standard $L$-smoothness assumption, and thus can be canceled out by the negative term in Eq. (7) by clipping the step size to $\eta_t \lesssim 1/L$. Therefore, most existing gradient-variation techniques require the prior knowledge of the smoothness parameter $L$.

Let us return to gradient-variation online learning with $(L_\nu, \nu)$-Hölder smoothness. Unfortunately we *cannot* directly apply the definition in Eq. (1) as we did with standard smoothness, because it would yield $\|\nabla f_{t-1}(\mathbf{x}_t) - \nabla f_{t-1}(\mathbf{x}_{t-1})\|^2 \leq L_\nu^2 \|\mathbf{x}_t - \mathbf{x}_{t-1}\|^{2\nu}$, which mismatches with the negative term. To this end, we present a key lemma regarding Hölder smoothness as a kind of *inexact* smoothness [Devolder et al., 2014], which has a similar form to standard smoothness except for an additional corruption term. The proof is in Appendix A.2.

**Lemma 1.** *Suppose the function $f$ is $(L_\nu, \nu)$-Hölder smooth. Then, for any $\delta > 0$, denoting by $L = \delta^{\frac{\nu-1}{1+\nu}} L_\nu^{\frac{2}{1+\nu}}$, it holds that for all $\mathbf{x}, \mathbf{y} \in \mathbb{R}^d$:*

$$\|\nabla f(\mathbf{x}) - \nabla f(\mathbf{y})\|^2 \leq L^2 \|\mathbf{x} - \mathbf{y}\|^2 + 4L\delta. \tag{8}$$

When smoothness holds, i.e., $\nu = 1$, Lemma 1 recovers the standard smoothness assumption when $\delta$ approaches 0. When functions are $G$-Lipschitz, i.e., $\nu = 0$ and $L_\nu = 2G$, by treating the right-hand side as a function for $\delta$ and calculating the minimum, the lemma results in $\|\nabla f_t(\mathbf{x}) - \nabla f_t(\mathbf{y})\|^2 \lesssim G^2$, providing an upper bound that depends only on $G$.

In the next step, applying Lemma 1 encounters another severe issue: the parameter $L$ in Lemma 1 is algorithmically *unavailable*, preventing us from explicitly setting the step size clipping $\eta_t \lesssim 1/L$. This is because the Hölder smoothness parameters $L_\nu$ and $\nu$ are unknown, and $\delta$ is chosen based on theoretical considerations and thus exists only in the analysis.

To handle this problem, inspired by Kavis et al. [2019], we adopt the following AdaGrad-style step sizes [Duchi et al., 2011] which allows us to perform *virtual* clipping in the analysis:

$$\eta_{t+1} \propto \frac{1}{\sqrt{A_t}}, \quad \text{where } A_t \triangleq \|\nabla f_1(\mathbf{x}_1)\|^2 + \sum_{s=2}^{t} \|\nabla f_s(\mathbf{x}_s) - M_s\|^2. \tag{9}$$

The rationale behind is that, since $\eta_{t+1}$ in Eq. (9) is non-increasing, it will eventually become smaller than $1/L$ after certain rounds, i.e., for $t > \tau$, thereby achieving implicit clipping. On the other hand, for $t \leq \tau$, the relation $\eta_{\tau+1} \propto 1/\sqrt{A_\tau} \gtrsim 1/L$ implies that $\sqrt{A_\tau}$ remains small. Hence, the uncancelled gradient-variation summation in Eq. (7), which is bounded by $\sqrt{A_\tau}$, is at most a constant.

Putting everything together, we establish the gradient-variation regret with the proof in Appendix A.3.

**Theorem 1.** *Consider online learning with convex and $(L_\nu, \nu)$-Hölder smooth functions. Under Assumption 1, optimistic OGD in Eq. (6) with $M_1 = \mathbf{0}, M_t = \nabla f_{t-1}(\mathbf{x}_{t-1})$ for all $t \geq 2$, and step sizes $\eta_t = \frac{D}{2\sqrt{A_{t-1}}}$ with $A_t$ defined in Eq. (9) for all $t \in [T]$, ensures the following regret bound:*

$$\text{REG}_T \leq \mathcal{O}\left( D\sqrt{V_T} + L_\nu D^{1+\nu} T^{\frac{1-\nu}{2}} + D\|\nabla f_1(\mathbf{x}_1)\| \right). \tag{10}$$

Theorem 1 implies optimal guarantees for both smooth and Lipschitz functions even in terms of the dependence on the domain diameter $D$: *(i)* when online functions are $L$-smooth, i.e., $(L, 1)$-Hölder smooth, our result recovers the optimal bound of $\mathcal{O}(D\sqrt{V_T} + LD^2)$ [Chiang et al., 2012]; and *(ii)* when online functions are $G$-Lipschitz, i.e., $(2G, 0)$-Hölder smooth, our result also recovers the worst-case minimax optimal guarantee $\mathcal{O}(GD\sqrt{T})$ [Zinkevich, 2003].

**Remark 1.** We emphasize that our algorithm is *strongly universal* (as defined in Definition 1), since it does *not* require knowledge of the Hölder smoothness parameters. In fact, even when restricted to gradient-variation online learning with smooth functions, our results imply an algorithm achieving an optimal $\mathcal{O}(D\sqrt{V_T} + LD^2)$ regret *without* requiring prior knowledge of the smoothness parameter $L$, unlike previous works that depend on it [Chiang et al., 2012; Yan et al., 2023; Zhao et al., 2024]. ◁

## 3.2 Implication to Offline Convex Optimization

In this section, we achieve acceleration for offline convex and $(L_\nu, \nu)$-Hölder smooth optimization in the stochastic setting, as defined in Section 2.1. This is accomplished by leveraging the effectiveness of the gradient-variation adaptivity presented in Section 3.1 and combining it with the stabilized online-to-batch conversion [Cutkosky, 2019]. The proof can be found in Appendix A.4.

**Theorem 2.** *Consider the optimization problem $\min_{\mathbf{x} \in \mathcal{X}} \ell(\mathbf{x})$ in the stochastic setting, where the objective $\ell$ is convex and $(L_\nu, \nu)$-Hölder smooth, under Assumption 1. Using the online-to-batch conversion (Algorithm 1) with weights $\alpha_t = t$ for all $t \in [T]$, and choosing the online algorithm $\mathcal{A}_{\mathrm{OL}}$ as optimistic OGD Eq. (6) with following configurations:*

- *setting the optimism as $M_1 = \mathbf{0}$, $M_t = \alpha_t \mathbf{g}(\widetilde{\mathbf{x}}_t)$ with $\widetilde{\mathbf{x}}_t = \frac{1}{\alpha_{1:t}}(\sum_{s=1}^{t-1} \alpha_s \mathbf{x}_s + \alpha_t \mathbf{x}_{t-1})$;*
- *setting the step size as $\eta_t = \frac{D}{2\sqrt{A_{t-1}}}$ with $A_t$ defined in Eq. (9).*

*Then we obtain the following last-iterate convergence rate for any $\mathbf{x}_\star \in \mathcal{X}$:*

$$\mathbb{E}\left[\ell(\overline{\mathbf{x}}_T)\right] - \ell(\mathbf{x}_\star) \leq \mathcal{O}\left(\frac{L_\nu D^{1+\nu}}{T^{\frac{1+3\nu}{2}}} + \frac{\sigma D}{\sqrt{T}} + \frac{D\|\nabla \ell(\mathbf{x}_1)\|}{T^2}\right).$$

Theorem 2 achieves *strong universality* due to its adaptivity to Hölder smoothness, matching the best-known result of Rodomanov et al. [2024], while our analysis is arguably much simpler due to explicitly decoupling the two algorithmic components — adaptive step sizes and gradient evaluation on weighted averaged iterates. For $L$-smooth and $G$-Lipschitz functions, our result recovers the optimal rates of $\mathcal{O}(LD^2/T^2 + \sigma D/\sqrt{T})$ and $\mathcal{O}((G + \sigma)D/\sqrt{T})$, respectively.

**Remark 2.** We have achieved strong universality in constrained stochastic optimization. However, the unconstrained setting presents additional challenges and remains less explored, especially with strong universality in unconstrained stochastic optimization still an open question [Rodomanov et al., 2024]. Although there have been some partial advancements in this area. In the deterministic setting, strong universality has been achieved: Orabona [2023] attained an $\mathcal{O}(L_\nu \|\mathbf{x}_\star\|^{1+\nu}/T^{(1+\nu)/2})$ rate, while Li and Lan [2025] obtained an accelerated $\mathcal{O}(L_\nu \|\mathbf{x}_\star\|^{1+\nu}/T^{(1+3\nu)/2})$ rate with the pre-specified accuracy. In the stochastic setting, progress has been limited to weak universality and sub-optimality [Ivgi et al., 2023; Kreisler et al., 2024]. To the best of our knowledge, achieving strong universality in unconstrained and stochastic optimization remains an open question. We leave the extension of our method to unconstrained optimization as an interesting future direction. ◁

## 4 Strongly Convex Optimization with Hölder Smoothness

This section focuses on strongly convex optimization with Hölder smoothness. Section 4.1 establishes gradient-variation regret bounds for online learning, Section 4.2 obtains a *weakly universal* method for offline optimization, and Section 4.3 develops an optimization algorithm that does not require the smoothness parameter or strong convexity curvature.

### 4.1 Gradient-Variation Online Strongly Convex Optimization with Hölder Smoothness

In this part, we study online optimization with strongly convex and Hölder smooth functions. In Theorem 3, we demonstrate that optimistic OGD, when properly configured, achieves the gradient-variation regret guarantee. The proof is provided in Appendix B.1.

**Theorem 3.** *Consider online learning with $\lambda$-strongly convex and $(L_\nu, \nu)$-Hölder smooth functions. Under Assumption 1, optimistic OGD in Eq. (6) with $M_1 = \mathbf{0}$, $M_t = \nabla f_{t-1}(\mathbf{x}_{t-1})$ for all $t \geq 2$, and step size $\eta_t = \frac{6}{\lambda t}$ for all $t \in [T]$, ensures the following regret bound:*

$$\mathrm{REG}_T \leq \mathcal{O}\left(\frac{\widehat{G}_{\max}^2}{\lambda} \log\left(1 + \frac{V_T}{\widehat{G}_{\max}^2}\right) + \frac{L_\nu^2 D^{2\nu}}{\lambda}(\log T)^{\frac{1-\nu}{1+\nu}} + \frac{\|\nabla f_1(\mathbf{x}_1)\|^2}{\lambda}\right),$$

*where $\widehat{G}_{\max}^2 \triangleq \max_{t \in [T-1]} \sup_{\mathbf{x} \in \mathcal{X}} \|\nabla f_t(\mathbf{x}) - \nabla f_{t+1}(\mathbf{x})\|^2$.*

Theorem 3 recovers best-known results under both smoothness and Lipschitzness: $\mathcal{O}(\frac{\widehat{G}_{\max}^2}{\lambda} \log(1 + V_T/\widehat{G}_{\max}^2) + \frac{1}{\lambda}L^2 D^2)$ for $L$-smooth functions [Chen et al., 2024] and $\mathcal{O}(\frac{G^2}{\lambda} \log T)$ for $G$-Lipschitz functions [Hazan et al., 2007; Abernethy et al., 2008], respectively.

## 4.2 Implication to Offline Strongly Convex Optimization

In this part, we develop a weakly universal algorithm for deterministic strongly convex optimization. This is done by leveraging the gradient-variation adaptivity with an online-to-batch conversion tailored for strongly convex optimization, and a carefully designed smoothness detection scheme.

We first introduce the motivation of our solution. As explained in Section 2.3, the online-to-batch conversion transforms the convergence rate into the regret divided by the total weight $\alpha_{1:T} = \sum_{t=1}^{T} \alpha_t$. To minimize regret, we employ an online algorithm with gradient-variation adaptivity, which leverages smoothness to convert the adaptivity term, allowing the positive term to be canceled out by the corresponding negative term. Now, let us consider the $\lambda$-strongly convex and $L_\ell$-smooth case. By tailoring an online-to-batch conversion specifically for strongly convex optimization, i.e., Lemma 5, the cancellation between the positive and negative terms hinges on analyzing the following expression:

$$\frac{2\alpha_t^2}{\lambda\alpha_{1:t}}\left\|\nabla\ell(\overline{\mathbf{x}}_t) - \nabla\ell(\overline{\mathbf{x}}_{t-1})\right\|^2 - \alpha_{1:t-1}\mathcal{D}_\ell(\overline{\mathbf{x}}_{t-1}, \overline{\mathbf{x}}_t), \tag{11}$$

If we directly use the smoothness property $\|\nabla\ell(\mathbf{x}) - \nabla\ell(\mathbf{y})\|^2 \leq 2L_\ell\mathcal{D}_\ell(\mathbf{y}, \mathbf{x})$ [Nesterov, 2018, Theorem 2.1.5] to bound the positive term, we would need $\alpha_t$ to satisfy $4\kappa\alpha_t^2 \leq \alpha_{1:t}\alpha_{1:t-1}$, where $\kappa \triangleq L_\ell/\lambda$. However, as we aim to design a universal algorithm that adapts to both smooth and non-smooth settings, the design of $\alpha_t$ must not rely on the smoothness parameter $L_\ell$.

Then, we design a novel smoothness-detection scheme. First, denoting the empirical smoothness parameter at the $t$-th iteration by $L_t \triangleq \frac{\|\nabla\ell(\overline{\mathbf{x}}_t) - \nabla\ell(\overline{\mathbf{x}}_{t-1})\|}{2\mathcal{D}_\ell(\overline{\mathbf{x}}_{t-1}, \overline{\mathbf{x}}_t)}$, where $L_t \leq L_\ell$, we proceed to analyze the cancellation between the following two terms:

$$\text{Eq. (11)} = \left(\frac{4\beta_t^2 L_t}{\lambda(1 + \beta_t)} - 1\right)\alpha_{1:t-1}\mathcal{D}_\ell(\overline{\mathbf{x}}_{t-1}, \overline{\mathbf{x}}_t),$$

where we define $\beta_t \triangleq \alpha_t/\alpha_{1:t-1}$ for simplicity. Ideally, the cancellation holds if $\beta_t \leq \sqrt{\lambda/(4L_t)}$. However, a challenge remains: $L_t$ is obtained only after $\beta_t$ has been determined. This arises from the use of optimistic OGD as the online algorithm in the online-to-batch conversion, requiring an additional update step that integrates $\beta_t$ information before computing $\mathbf{x}_t$ and consequently $L_t$.

To this end, we designed a method that first guesses a $\beta_t$, and then decides whether to adjust the guess based on the observed $L_t$. Specifically, if the guessed $\beta_t$ fails to meet the requirements $\beta_t \leq \sqrt{\lambda/(4L_t)}$, we discard the current $\mathbf{x}_t$, halve $\beta_t$, and recompute $\mathbf{x}_t$. We then repeat this guess-and-check procedure until the requirement is satisfied. As long as we can ensure a reasonable lower bound for $\beta_t$, the number of wasted updates will be logarithmic, which will only add a multiplicative constant factor to the final bound. The simplest design is to explicitly define a lower bound $\bar{\beta}$ for $\beta_t$, which acts as a safeguard to guarantee a convergence rate in non-smooth scenarios. For the $L_\ell$-smooth case, our mechanism implicitly provides an adaptive lower bound $\frac{1}{2}\sqrt{\lambda/(4L_\ell)}$. This arises from the fact that when $\beta_t \leq \sqrt{\lambda/(4L_\ell)}$, we directly obtain $\beta_t \leq \sqrt{\lambda/(4L_t)}$ since $L_\ell \geq L_t$. In this case, $\beta_t$ will no longer be decreased.

To conclude, there are three key ingredients in our solution: *(i)* online-to-batch conversion tailored for strongly convex optimization (i.e., Lemma 5), *(ii)* the guess-and-check smoothness detection scheme, and *(iii)* a *one-step* variant of optimistic OGD as the online algorithm (i.e., Lemma 11). We provide the convergence guarantee in Theorem 4 with the proof in Appendix B.3.

**Theorem 4.** *Consider the optimization problem $\min_{\mathbf{x}\in\mathcal{X}} \ell(\mathbf{x})$ in the deterministic setting, where the objective $\ell$ is $\lambda$-strongly convex and $G$-Lipschitz. Then Algorithm 2 with $\beta_1 = 1, \bar{\beta} = \exp(\frac{1}{T}\ln T) - 1$ ensures that*

$$\ell(\overline{\mathbf{x}}_\tau) - \ell(\mathbf{x}_\star) \leq \mathcal{O}\left(\frac{G^2}{\lambda}\min\left\{\exp\left(\frac{-T}{6\sqrt{\kappa}}\right), \frac{\log T}{T}\right\}\right),$$

*without the knowledge of $G$ or the smoothness parameter $L_\ell$, where $\kappa \triangleq L_\ell/\lambda$ denotes the condition number, and we define $L_\ell \triangleq \infty$ if $\ell$ is non-smooth.*

Theorem 4 demonstrates the *weak universality* of Algorithm 2, meaning that it maintains the respective *near-optimal* convergence rates in both smooth and non-smooth cases, without knowledge of the parameters $L_\ell$ or $G$. However, a slight issue arises similar to that in Levy [2017]: to achieve universality, both our method and theirs depend on the Lipschitz continuity of $\ell$, even though the

---

**Algorithm 2** Universal Accelerated Strongly Convex Optimization

---

**Input:** Strong convexity curvature $\lambda$, $\beta_1$ and threshold $\bar{\beta}$, oracle queries budget $T$ and $\mathbf{x}_1 \in \mathcal{X}$.

1: **Initialization:** $\alpha_1 = 1, \overline{\mathbf{x}}_1 = \mathbf{x}_1, M_1 = \mathbf{0}$, index $t = 1$, oracle queries count $c = 1$.
2: **while** $c < T$ **do**
3:     Construct $\mathbf{g}_t = \alpha_t \nabla \ell(\overline{\mathbf{x}}_t) + \lambda \alpha_t(\mathbf{x}_t - \overline{\mathbf{x}}_t)$, set $\beta_{t+1} = \beta_t$
4:     **while** $c < T$ **do**
5:        Set $\alpha_{t+1} = \beta_{t+1}\alpha_{1:t}$, calculate $\widetilde{\mathbf{x}}_{t+1} = \frac{1}{\alpha_{1:t+1}}(\alpha_{1:t}\overline{\mathbf{x}}_t + \alpha_{t+1}\mathbf{x}_t)$      ▷ Guess procedure
6:        Construct $M_{t+1} = \alpha_{t+1}\nabla \ell(\overline{\mathbf{x}}_t) + \lambda\alpha_{t+1}(\mathbf{x}_t - \widetilde{\mathbf{x}}_{t+1})$
7:        Update $\mathbf{x}_{t+1} = \Pi_{\mathcal{X}}[\mathbf{x}_t - \eta_t(\mathbf{g}_t - M_t + M_{t+1})]$ with $\eta_t = \frac{1}{\lambda\alpha_{1:t}}$
8:        Calculate $\overline{\mathbf{x}}_{t+1} = \frac{1}{\alpha_{1:t+1}}(\alpha_{1:t}\overline{\mathbf{x}}_t + \alpha_{t+1}\mathbf{x}_{t+1})$, query $\nabla \ell(\overline{\mathbf{x}}_{t+1})$, count $c \leftarrow c + 1$
9:        **if** $\beta_{t+1} = \bar{\beta}$ **then:** $t \leftarrow t + 1$, **break**
10:       Calculate $L_{t+1} \triangleq \frac{\|\nabla \ell(\overline{\mathbf{x}}_t) - \nabla \ell(\overline{\mathbf{x}}_{t+1})\|^2}{2D_\ell(\overline{\mathbf{x}}_t, \overline{\mathbf{x}}_{t+1})}$      ▷ Check procedure
11:       **if** $\beta_{t+1} \leq \sqrt{\frac{\lambda}{4L_{t+1}}}$ **then:** $t \leftarrow t + 1$, **break**
12:       **else** $\beta_{t+1} = \max\{\frac{\beta_{t+1}}{2}, \bar{\beta}\}$

**Output:** $\overline{\mathbf{x}}_\tau$ with $\tau = t$ the final iteration.

---

specific parameter is not required. We conjecture that Lipschitz continuity might be a necessary condition for universality in strongly convex optimization. Further investigation is needed.

Additionally, our algorithm Algorithm 2 is highly flexible and can achieve better theoretical guarantees when more information about smoothness is available. For further details, see Corollary 1.

**Remark 3.** To the best of our knowledge, Levy [2017] achieved the previously best-known universal results for strongly convex optimization, in which an adaptive normalized gradient descent is employed with online-to-batch conversion weights inversely proportional to the square of the gradient norm. In the deterministic setup, the author achieved an $\mathcal{O}((\log T)/T)$ convergence rate for the Lipschitz function, and an $\mathcal{O}(\exp(-T/\kappa) \cdot T/\kappa)$ rate for smooth and Lipschitz objectives. Our work improves upon their result by designing a weakly universal algorithm with the *first* accelerated rate of $\mathcal{O}(\exp(-T/(6\sqrt{\kappa})))$ for smooth and Lipschitz functions. However, our method relies on a smoothness detection scheme based on the observed gradients, which only works in the deterministic setting for now. Extending it to the stochastic setting remains challenging. ◁

**Remark 4.** Designing a *strongly universal*, i.e., adapting to Hölder smoothness, method for strongly convex optimization is still an open problem. Notably, given the Hölder smoothness parameters, Devolder et al. [2013] have established a sample-complexity-based rate that can recover the (near-)optimal rate for smooth and non-smooth cases, which may serve as a starting point. ◁

## 4.3   Grid Search for the Unknown Strong Convexity Curvature

Algorithm 2 shows strong adaptivity to the unknown smoothness parameter $L_\ell$, and in this part, we further enhance its adaptivity by removing the strong convexity curvature $\lambda$ as the algorithmic input.[2]

We consider the strongly convex optimization $\min_{\mathbf{x} \in \mathbb{R}^d} \ell(\mathbf{x})$ in the deterministic setting, where $\ell(\mathbf{x})$ is $L_\ell$-smooth and $\lambda$-strongly convex, but the algorithm does not know $L_\ell$ and $\lambda$.

For this setting, the best-known result is achieved by Lan et al. [2023], who obtained the optimal sample complexity with a pre-specified target error $\varepsilon$. However, their sample complexity bound, when translated into a convergence rate for the sub-optimality gap, is expressed as $\mathcal{O}(\exp(-T/(882\sqrt{\kappa})))$ and thus not optimal (see further details in Remark 6). Whereas we design an algorithm achieving an $\exp(-T/((1 + 4\sqrt{2\kappa})\lceil 2\log_2 T\rceil))$ convergence rate, with only the oracle queries budget $T$ as input.

Algorithm 3 outlines the main procedures. Essentially, it runs multiple instances of Algorithm 2 to search for the strong convexity parameter $\lambda$ by selecting the output with the smallest loss. Notably, a proper choice of the search range for $\lambda$ is critical for success. In our algorithm, this range is derived

---

[2]In online learning, adapting to unknown curvature is known as "universal online learning", where a widely adopted technique is to run multiple base algorithms for exploration and use a meta algorithm for exploitation.

---

**Algorithm 3** Universal Accelerated Strongly Convex Optimization, Search Method

---

**Input:** Total oracle queries budget $T$.
1: **Initialization:** $M = \lceil 2 \log_2 T \rceil, \mathbf{x}^0 \in \mathcal{X} = \mathbb{R}^d$ and $\widehat{\lambda} = \frac{\|\nabla \ell(\mathbf{a}) - \nabla \ell(\mathbf{b})\|}{\|\mathbf{a} - \mathbf{b}\|}$ with any $\mathbf{a}, \mathbf{b} \in \mathbb{R}^d$.
2: **for** $i = 1, 2, \ldots, M$ **do**
3:   Run Algorithm 2 with $\left( \lambda_i = 2^{-i} \cdot \widehat{\lambda}, \beta_1 = 1, \bar{\beta} = 0, T_i = \frac{T}{M}, \mathbf{x}_1 = \mathbf{x}^0 \right)$, receive $\mathbf{x}^i$.
4: **end for**
**Output:** $\mathbf{x}^{i_\star}$ with $i_\star = \arg\min_{0 \leq i \leq M} \{\ell(\mathbf{x}^i)\}$.

---

through rigorous analysis by carefully exploiting properties of smoothness and strong convexity, rather than imposing assumptions about the upper or lower bounds of $\lambda$. The following theorem provides the convergence rate, with the proof provided in Appendix B.4.

**Theorem 5.** *Consider the optimization problem* $\min_{\mathbf{x} \in \mathbb{R}^d} \ell(\mathbf{x})$ *in the deterministic setting, where* $\ell(\cdot)$ *is* $\lambda$-*strongly convex and* $L_\ell$-*smooth. Denoted by* $\kappa \triangleq L_\ell/\lambda$. *Then,* Algorithm 3 *guarantees*

$$\ell(\mathbf{x}^{i_\star}) - \ell(\mathbf{x}_\star) \leq \mathcal{O}\left( \frac{\|\nabla \ell(\mathbf{x}_1)\|^2}{\lambda} \exp\left( \frac{-T}{(1 + 4\sqrt{2\kappa})\lceil 2 \log_2 T \rceil} \right) \right),$$

*which is achieved without the knowledge of* $L_\ell$ *and* $\lambda$.

**Remark 5.** The limitation of both Theorem 5 and Lan et al. [2023] is that neither algorithm can guarantee convergence in the non-smooth case, i.e., when $L_\ell = \infty$. However, our result has an advantage in terms of the convergence rate. The result of Lan et al. [2023], when translated into the convergence rate for the sub-optimality gap, is expressed as $\mathcal{O}(\exp(-T/(882\sqrt{\kappa})))$, with a notably large denominator 882 in the exponent. Consequently, despite the $\log T$ factor in our Theorem 5, it remains highly competitive and even surpasses Lan et al. [2023] when $T \leq 8.7 \times 10^{19}$. Further details can be found in Appendix B.4.

**Remark 6.** We note that when expressing exponential convergence, the use of asymptotic notation differs between convergence rate and sample complexity. To understand this, let us reconsider the sample complexity $T \leq \alpha \log(\beta/\varepsilon) = \mathcal{O}(\log(\beta/\varepsilon))$ required to achieve the target error $\varepsilon$ and the corresponding convergence rate $\varepsilon \leq \beta \exp(-T/\alpha) = \mathcal{O}(\exp(-T/\alpha))$, where $\alpha, \beta$ are two constants. It can be observed that the constant $\alpha$ in the asymptotic notation for sample complexity has an *exponential* impact on the convergence rate. In contrast, the constant $\beta$ in the asymptotic bound of the convergence rate influences the sample complexity only *logarithmically*. Thus in this case, achieving optimal sample complexity does not necessarily guarantee optimal convergence rate.

## 5  Conclusion

In this work, we explore gradient-variation online learning with Hölder smoothness and its implications to offline optimization. For online learning with Hölder smoothness, we establish the first gradient-variation regret bounds for (strongly) convex online functions, seamlessly interpolating between the optimal regret rates in the smooth and non-smooth regimes. For offline optimization, we develop a series of universal optimization methods by leveraging gradient-variation online adaptivity, stabilized online-to-batch conversion, and carefully designed components such as detection-based procedures and grid search tailored specifically for strongly convex cases. Our convergence rates match the existing optimal universal results for convex optimization and significantly improve upon non-accelerated rates for strongly convex optimization.

An important open problem is designing gradient-variation online adaptivity and extending its implications to offline optimization in the *unconstrained* setting. Another interesting direction is to further develop offline optimization algorithms by leveraging insights from adaptive online learning.

## Acknowledgments

This research was supported by NSFC (62361146852).

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

# A  Omitted Details for Section 3

In this section, we first provide some useful lemmas for Hölder smoothness, then give the proofs of theorems in Section 3.

## A.1  Useful Lemmas for Hölder Smoothness

This part provides several useful lemmas for Hölder smoothness.

**Lemma 2** (Lemma 1 of Nesterov [2015]). *Let convex function $f : \mathcal{X} \to \mathbb{R}$ over the convex set $\mathcal{X}$ be $(L_\nu, \nu)$-Hölder smooth.[3] Then for any $\delta > 0$, denoting by $L = \delta^{\frac{\nu-1}{1+\nu}} L_\nu^{\frac{2}{1+\nu}}$, for all $\mathbf{x}, \mathbf{y} \in \mathcal{X}$:*

$$f(\mathbf{y}) - f(\mathbf{x}) - \langle \nabla f(\mathbf{x}), \mathbf{y} - \mathbf{x} \rangle \leq \frac{L}{2} \|\mathbf{x} - \mathbf{y}\|^2 + \delta. \tag{12}$$

**Lemma 3** (Theorem 1 of Devolder et al. [2014]). *If convex function $f : \mathcal{X} \to \mathbb{R}$ over the convex set $\mathcal{X}$ satisfies that, there exists positive constants $L$ and $\delta$ such that, for all $\mathbf{x}, \mathbf{y} \in \mathcal{X}$:*

$$f(\mathbf{y}) - f(\mathbf{x}) - \langle \nabla f(\mathbf{x}), \mathbf{y} - \mathbf{x} \rangle \leq \frac{L}{2} \|\mathbf{x} - \mathbf{y}\|^2 + \delta, \tag{13}$$

*then for all $\mathbf{x}, \mathbf{y} \in \mathcal{X}$:*

$$\frac{1}{2L} \|\nabla f(\mathbf{x}) - \nabla f(\mathbf{y})\|^2 \leq f(\mathbf{y}) - f(\mathbf{x}) - \langle \nabla f(\mathbf{x}), \mathbf{y} - \mathbf{x} \rangle + \delta. \tag{14}$$

**Lemma 4** (Theorem A.2. of Rodomanov et al. [2024]). *If convex function $f : \mathbb{R}^d \to \mathbb{R}$ over $\mathbb{R}^d$ satisfies that, there exists positive constants $L$ and $\delta$ such that, for all $\mathbf{x}, \mathbf{y} \in \mathbb{R}^d$:*

$$f(\mathbf{y}) - f(\mathbf{x}) - \langle \nabla f(\mathbf{x}), \mathbf{y} - \mathbf{x} \rangle \leq \frac{L}{2} \|\mathbf{x} - \mathbf{y}\|^2 + \delta, \tag{15}$$

*then for all $\mathbf{x}, \mathbf{y} \in \mathbb{R}^d$:*

$$\|\nabla f(\mathbf{x}) - \nabla f(\mathbf{y})\|^2 \leq 2L \mathcal{D}_f(\mathbf{x}, \mathbf{y}) + 2L\delta. \tag{16}$$

## A.2  Proof of Lemma 1

*Proof.* Since $f$ is $(L_\nu, \nu)$-Hölder smooth, by combining Lemma 2 and Lemma 3, for any $\delta > 0$, denoting by $L = \delta^{\frac{\nu-1}{1+\nu}} L_\nu^{\frac{2}{1+\nu}}$, for all $\mathbf{x}, \mathbf{y} \in \mathcal{X}$:

$$\frac{1}{2L} \|\nabla f(\mathbf{x}) - \nabla f(\mathbf{y})\|^2 \overset{(14)}{\leq} f(\mathbf{y}) - f(\mathbf{x}) - \langle \nabla f(\mathbf{x}), \mathbf{y} - \mathbf{x} \rangle + \delta \overset{(12)}{\leq} \frac{L}{2} \|\mathbf{x} - \mathbf{y}\|^2 + 2\delta. \tag{17}$$

Multiplying both sides of the inequality by $2L$ completes the proof. $\qquad\square$

## A.3  Proof of Theorem 1

*Proof.* Applying Lemma 10 with comparators $\mathbf{u}_t = \mathbf{x}_\star = \arg\min_{\mathbf{x} \in \mathcal{X}} \sum_{t=1}^{T} f_t(\mathbf{x})$ for all $t \in [T]$,

$$
\begin{aligned}
\text{REG}_T = \sum_{t=1}^{T} f_t(\mathbf{x}_t) - \sum_{t=1}^{T} f_t(\mathbf{x}_\star) &\leq \sum_{t=1}^{T} \langle \nabla f_t(\mathbf{x}_t), \mathbf{x}_t - \mathbf{x}_\star \rangle \\
&\leq \sum_{t=1}^{T} \eta_{t+1} \|\nabla f_t(\mathbf{x}_t) - M_t\|^2 + \frac{D^2}{\eta_{T+1}} - \sum_{t=2}^{T} \frac{1}{8\eta_{t+1}} \|\mathbf{x}_t - \mathbf{x}_{t-1}\|^2 \\
&\leq 3D\sqrt{A_T} - \sum_{t=2}^{T} \frac{1}{8\eta_{t+1}} \|\mathbf{x}_t - \mathbf{x}_{t-1}\|^2,
\end{aligned}
\tag{18}
$$

where $A_t \triangleq \|\nabla f_1(\mathbf{x}_1)\|^2 + \sum_{s=2}^{t} \|\nabla f_s(\mathbf{x}_s) - M_s\|^2$, and we apply Lemma 12 in the last line.

---

[3]Though $\mathcal{X}$ is supposed to be closed in Nesterov [2015], this lemma holds for $\mathcal{X} = \mathbb{R}^d$ with the same proof.

If $\sqrt{A_T} \leq 2LD$, we finish the proof trivially, so in the following, we assume $\sqrt{A_T} > 2LD$.

Define $t_0$ that, if $\sqrt{A_1} > 2LD$, let $t_0 = 1$, otherwise let $t_0 = \min\{t : t \in [T-1], \sqrt{A_{t+1}} > 2LD\}$. Then we have $\sqrt{A_{t_0}} \leq \|\nabla f_1(\mathbf{x}_1)\| + 2LD$, while for all $t_0 + 1 \leq t \leq T$ it holds that $\sqrt{A_t} > 2LD$.

Because all online functions are $(L_\nu, \nu)$-Hölder smooth and applying Lemma 1, we show the following decomposition for $\alpha\sqrt{A_T}$ with constant $\alpha > 0$. For any $\delta > 0$ that only exists in analysis, denoting by $L = \delta^{\frac{\nu-1}{1+\nu}} L_\nu^{\frac{2}{1+\nu}}$:

$$\alpha\sqrt{A_T} \leq \alpha\sqrt{A_{t_0}} + \alpha\sqrt{\sum_{t=t_0+1}^{T} \|\nabla f_t(\mathbf{x}_t) - \nabla f_{t-1}(\mathbf{x}_t) + \nabla f_{t-1}(\mathbf{x}_t) - \nabla f_{t-1}(\mathbf{x}_{t-1})\|_*^2}$$

$$\leq \alpha\sqrt{A_{t_0}} + \alpha\sqrt{2V_T} + \alpha\sqrt{2L^2 \sum_{t=t_0+1}^{T} \|\mathbf{x}_t - \mathbf{x}_{t-1}\|^2 + 8L \sum_{t=t_0+1}^{T} \delta}$$

$$\leq \alpha\sqrt{A_{t_0}} + \alpha\sqrt{2V_T} + \alpha^2 L + \frac{L}{2} \sum_{t=t_0+1}^{T} \|\mathbf{x}_t - \mathbf{x}_{t-1}\|^2 + \alpha\sqrt{8L\delta T}.$$

With this decomposition, we prove the regret bound in the following with $\alpha = 3D$:

$$\text{REG}_T \leq 3D\sqrt{A_{t_0}} + 3D\sqrt{2V_T} + 9LD^2 + \sum_{t=t_0+1}^{T} \left(\frac{L}{2} - \frac{1}{8\eta_{t+1}}\right) \|\mathbf{x}_t - \mathbf{x}_{t-1}\|^2 + 3D\sqrt{8L\delta T}$$

$$\leq 3D\sqrt{2V_T} + 15LD^2 + 3D\|\nabla f_1(\mathbf{x}_1)\| + 3D\sqrt{8L\delta T}.$$

Then by choosing $\delta = L_\nu D^{1+\nu} T^{-\frac{1+\nu}{2}}$ (that only exists in analysis), we obtain

$$\text{REG}_T \leq \mathcal{O}\left(D\sqrt{V_T} + L_\nu D^{1+\nu} T^{\frac{1-\nu}{2}} + D\|\nabla f_1(\mathbf{x}_1)\|\right),$$

which completes the proof. $\qquad\square$

### A.4 Proof of Theorem 2

*Proof.* With optimistic OGD as the online algorithm, by defining $f_t(\mathbf{x}) \triangleq \langle \alpha_t \mathbf{g}(\overline{\mathbf{x}}_t), \mathbf{x}\rangle$, we have:

$$\sum_{t=1}^{T} \alpha_t \langle \mathbf{g}(\overline{\mathbf{x}}_t), \mathbf{x}_t - \mathbf{x}_\star\rangle = \sum_{t=1}^{T} f_t(\mathbf{x}_t) - \sum_{t=1}^{T} f_t(\mathbf{x}_\star) \overset{(18)}{\leq} 3D\sqrt{A_T} - \sum_{t=2}^{T} \frac{1}{8\eta_{t+1}} \|\mathbf{x}_t - \mathbf{x}_{t-1}\|^2.$$

Now we trivially assume $\sqrt{A_T} > 4LD$, and define $t_0 \in [T-1]$ that, if $\sqrt{A_1} > 4LD$, let $t_0 = 1$, otherwise let $t_0 = \min\{t : t \in [T-1], \sqrt{A_{t+1}} > 4LD\}$. Then we have $\sqrt{A_{t_0}} \leq \|\nabla f_1(\mathbf{x}_1)\| + 4LD$, while for all $t_0 + 1 \leq t \leq T$ it holds that $\sqrt{A_t} > 4LD$. Continuing with our previous inequality:

$$\sum_{t=1}^{T} \alpha_t \langle \mathbf{g}(\overline{\mathbf{x}}_t), \mathbf{x}_t - \mathbf{x}_\star\rangle$$

$$\leq 3D\sqrt{A_{t_0}} + 3D\sqrt{\sum_{t=t_0+1}^{T} \alpha_t^2 \|\mathbf{g}(\overline{\mathbf{x}}_t) - \mathbf{g}(\widetilde{\mathbf{x}}_t)\|^2 - \sum_{t=2}^{T} \frac{1}{8\eta_{t+1}} \|\mathbf{x}_t - \mathbf{x}_{t-1}\|^2}$$

$$\leq 3D\sqrt{A_{t_0}} + 3D\sqrt{\sum_{t=t_0+1}^{T} 3\alpha_t^2 \|\nabla\ell(\overline{\mathbf{x}}_t) - \nabla\ell(\widetilde{\mathbf{x}}_t)\|^2 - \sum_{t=2}^{T} \frac{1}{8\eta_{t+1}} \|\mathbf{x}_t - \mathbf{x}_{t-1}\|^2}$$

$$+ 3D\sqrt{\sum_{t=t_0+1}^{T} 3\alpha_t^2 \|\nabla\ell(\overline{\mathbf{x}}_t) - \mathbf{g}(\overline{\mathbf{x}}_t)\|^2} + 3D\sqrt{\sum_{t=t_0+1}^{T} 3\alpha_t^2 \|\nabla\ell(\widetilde{\mathbf{x}}_t) - \mathbf{g}(\widetilde{\mathbf{x}}_t)\|^2},$$

where we use $\|\mathbf{a} + \mathbf{b} + \mathbf{c}\|^2 \leq 3\|\mathbf{a}\|^2 + 3\|\mathbf{b}\|^2 + 3\|\mathbf{c}\|^2$ for any $\mathbf{a}, \mathbf{b}, \mathbf{c} \in \mathbb{R}^d$. Now by taking expectation and using Jensen's inequality, i.e., $\mathbb{E}_x[\sqrt{x}] \leq \sqrt{\mathbb{E}_x[x]}$, we have

$$
\mathbb{E}\left[\sum_{t=1}^{T} \alpha_t \langle \mathbf{g}(\overline{\mathbf{x}}_t), \mathbf{x}_t - \mathbf{x}_\star \rangle\right]
$$

$$
\leq \mathbb{E}\left[3D\sqrt{\sum_{t=t_0+1}^{T} 3\alpha_t^2 \|\nabla\ell(\overline{\mathbf{x}}_t) - \nabla\ell(\widetilde{\mathbf{x}}_t)\|^2 - \sum_{t=2}^{T} \frac{1}{8\eta_{t+1}}\|\mathbf{x}_t - \mathbf{x}_{t-1}\|^2}\right]
$$

$$
+ 3D\|\nabla\ell(\mathbf{x}_1)\| + 12LD^2 + 12\sqrt{2}\sigma DT^{\frac{3}{2}},
$$

where we apply $\mathbb{E}[\|\mathbf{g}(\mathbf{x}) - \nabla\ell(\mathbf{x})\|^2 \mid \mathbf{x}] \leq \sigma^2$. By Lemma 1 and the definitions of $\overline{\mathbf{x}}_t, \widetilde{\mathbf{x}}_t$,

$$
\alpha_t^2\|\nabla\ell(\overline{\mathbf{x}}_t) - \nabla\ell(\widetilde{\mathbf{x}}_t)\|^2 \overset{(8)}{\leq} \alpha_t^2 L^2 \|\overline{\mathbf{x}}_t - \widetilde{\mathbf{x}}_t\|^2 + 4\alpha_t^2 L\delta = \frac{\alpha_t^4 L^2}{\alpha_{1:t}^2}\|\mathbf{x}_t - \mathbf{x}_{t-1}\|^2 + 4\alpha_t^2 L\delta
$$

$$
\leq 4L^2\|\mathbf{x}_t - \mathbf{x}_{t-1}\|^2 + 4t^2 L\delta.
$$

Then we have

$$
3D\sqrt{\sum_{t=t_0+1}^{T} 3\alpha_t^2 \|\nabla\ell(\overline{\mathbf{x}}_t) - \nabla\ell(\widetilde{\mathbf{x}}_t)\|^2 - \sum_{t=2}^{T} \frac{1}{8\eta_{t+1}}\|\mathbf{x}_t - \mathbf{x}_{t-1}\|^2}
$$

$$
\leq 6D\sqrt{3L^2\sum_{t=t_0+1}^{T}\|\mathbf{x}_t - \mathbf{x}_{t-1}\|^2 - \sum_{t=2}^{T}\frac{1}{8\eta_{t+1}}\|\mathbf{x}_t - \mathbf{x}_{t-1}\|^2} + 12\sqrt{2}D\sqrt{L\delta}T^{\frac{3}{2}}
$$

$$
\leq 27LD^2 + \sum_{t=t_0+1}^{T}\left(L - \frac{1}{8\eta_{t+1}}\right)\|\mathbf{x}_t - \mathbf{x}_{t-1}\|^2 + 12\sqrt{2}D\sqrt{L\delta}T^{\frac{3}{2}}
$$

$$
\leq 27LD^2 + 12\sqrt{2}D\sqrt{L\delta}T^{\frac{3}{2}}.
$$

Therefore, by combining the above inequalities we obtain

$$
\mathbb{E}\left[\ell(\overline{\mathbf{x}}_T)\right] - \ell(\mathbf{x}_\star) \leq \frac{1}{\alpha_{1:T}}\mathbb{E}\left[\sum_{t=1}^{T}\alpha_t\langle\mathbf{g}(\overline{\mathbf{x}}_t), \mathbf{x}_t - \mathbf{x}_\star\rangle\right]
$$

$$
\leq \frac{6D\|\nabla\ell(\mathbf{x}_1)\| + 78LD^2}{T^2} + \frac{24\sqrt{2}D\sqrt{L\delta} + 24\sqrt{2}\sigma D}{\sqrt{T}}.
$$

Then by setting $\delta = L_\nu D^{1+\nu}T^{\frac{-(3+3\nu)}{2}}$, we achieve the convergence rate of

$$
\mathbb{E}\left[\ell(\overline{\mathbf{x}}_T)\right] - \ell(\mathbf{x}_\star) \leq \mathcal{O}\left(\frac{L_\nu D^{1+\nu}}{T^{\frac{1+3\nu}{2}}} + \frac{\sigma D}{\sqrt{T}} + \frac{D\|\nabla\ell(\mathbf{x}_1)\|}{T^2}\right),
$$

which completes the proof. $\qquad\square$

## B  Omitted Details for Section 4

In this section, we give the proofs of theorems in Section 4.

### B.1  Proof of Theorem 3

*Proof.* We apply Lemma 9 with $\mathbf{u}_t = \mathbf{x}_\star = \arg\min_{\mathbf{x}\in\mathcal{X}}\sum_{t=1}^{T}f_t(\mathbf{x})$:

$$
\text{REG}_T = \sum_{t=1}^{T}f_t(\mathbf{x}_t) - \sum_{t=1}^{T}f_t(\mathbf{x}_\star) = \sum_{t=1}^{T}\langle\nabla f_t(\mathbf{x}_t), \mathbf{x}_t - \mathbf{x}_\star\rangle - \sum_{t=1}^{T}\mathcal{D}_{f_t}(\mathbf{x}_\star, \mathbf{x}_t)
$$

$$
\leq \sum_{t=1}^{T}\langle\nabla f_t(\mathbf{x}_t), \mathbf{x}_t - \mathbf{x}_\star\rangle - \frac{\lambda}{4}\sum_{t=1}^{T}\|\mathbf{x}_\star - \mathbf{x}_t\|^2 - \frac{1}{2}\sum_{t=1}^{T}\mathcal{D}_{f_t}(\mathbf{x}_\star, \mathbf{x}_t)
$$

$$\overset{(23)}{\leq} \underbrace{\sum_{t=1}^{T} \frac{1}{2\eta_t}\left(\|\mathbf{x}_\star - \widehat{\mathbf{x}}_t\|^2 - \|\mathbf{x}_\star - \widehat{\mathbf{x}}_{t+1}\|^2\right) - \frac{\lambda}{4}\sum_{t=1}^{T}\|\mathbf{x}_\star - \mathbf{x}_t\|^2}_{\text{TERM-A}}$$

$$+ \underbrace{\sum_{t=1}^{T}\eta_t\|\nabla f_t(\mathbf{x}_t) - \nabla f_{t-1}(\mathbf{x}_{t-1})\|^2}_{\text{TERM-B}} - \underbrace{\frac{1}{2}\sum_{t=1}^{T}\mathcal{D}_{f_t}(\mathbf{x}_\star, \mathbf{x}_t)}_{\text{TERM-C}}.$$

In the second line above, we use $\mathcal{D}_{f_t}(\mathbf{x}, \mathbf{y}) \geq \frac{\lambda}{2}\|\mathbf{x} - \mathbf{y}\|^2$ by the $\lambda$-strong convexity of $f_t$. We first investigate TERM-A. Since $\eta_t = \frac{6}{\lambda t}$,

$$\text{TERM-A} \leq \frac{\|\mathbf{x}_\star - \widehat{\mathbf{x}}_1\|^2}{2\eta_1} + \sum_{t=2}^{T}\left(\frac{1}{2\eta_t} - \frac{1}{2\eta_{t-1}}\right)\|\mathbf{x}_\star - \widehat{\mathbf{x}}_t\|^2 - \frac{\lambda}{4}\sum_{t=1}^{T}\|\mathbf{x}_\star - \mathbf{x}_t\|^2$$

$$\leq \frac{\lambda}{12}\sum_{t=1}^{T-1}\left(\|\mathbf{x}_\star - \widehat{\mathbf{x}}_{t+1}\|^2 - 2\|\mathbf{x}_\star - \mathbf{x}_t\|^2\right) \leq \frac{\lambda}{6}\sum_{t=1}^{T-1}\|\mathbf{x}_t - \widehat{\mathbf{x}}_{t+1}\|^2$$

$$\leq \frac{\lambda}{6}\sum_{t=1}^{T-1}\eta_t^2\|\nabla f_t(\mathbf{x}_t) - \nabla f_{t-1}(\mathbf{x}_{t-1})\|^2 \leq \text{TERM-B},$$

where in the second line we use $\widehat{\mathbf{x}}_1 = \mathbf{x}_1$. And in the last line above we apply Lemma 7 [Chiang et al., 2012]. Then by combining TERM-A, TERM-B and TERM-C together and applying Lemma 4 with arbitrary $\delta > 0$ that only exists in analysis, and denoting by $L = \delta^{\frac{\nu-1}{1+\nu}}L_\nu^{\frac{2}{1+\nu}}$, we obtain:

$$\text{REG}_T \leq \sum_{t=1}^{T}\frac{12}{\lambda t}\|\nabla f_t(\mathbf{x}_t) - \nabla f_{t-1}(\mathbf{x}_{t-1})\|^2 - \frac{1}{2}\sum_{t=1}^{T}\mathcal{D}_{f_t}(\mathbf{x}_\star, \mathbf{x}_t)$$

$$\leq \sum_{t=1}^{T}\frac{12}{\lambda t}\|\nabla f_t(\mathbf{x}_t) - \nabla f_t(\mathbf{x}_\star) + \nabla f_t(\mathbf{x}_\star) - \nabla f_{t-1}(\mathbf{x}_\star) + \nabla f_{t-1}(\mathbf{x}_\star) - \nabla f_{t-1}(\mathbf{x}_{t-1})\|^2$$

$$- \frac{1}{2}\sum_{t=1}^{T}\mathcal{D}_{f_t}(\mathbf{x}_\star, \mathbf{x}_t)$$

$$\leq \sum_{t=1}^{T}\frac{36}{\lambda t}\|\nabla f_t(\mathbf{x}_\star) - \nabla f_{t-1}(\mathbf{x}_\star)\|^2 + \sum_{t=1}^{T}\left(\frac{144L}{\lambda t} - \frac{1}{2}\right)\mathcal{D}_{f_t}(\mathbf{x}_\star, \mathbf{x}_t) + \sum_{t=1}^{T}\frac{144L\delta}{\lambda t}$$

$$\leq \sum_{t=1}^{T}\frac{36}{\lambda t}\sup_{\mathbf{x}\in\mathcal{X}}\|\nabla f_t(\mathbf{x}) - \nabla f_{t-1}(\mathbf{x})\|^2 + \sum_{t=1}^{T}\left(\frac{144L}{\lambda t} - \frac{1}{2}\right)\mathcal{D}_{f_t}(\mathbf{x}_\star, \mathbf{x}_t) + \frac{144L\delta(1+\ln T)}{\lambda}.$$

The first two terms can be well controlled by two technical lemmas (Lemma 13, Lemma 14), hence:

$$\text{REG}_T \leq \frac{36\widehat{G}_{\max}^2}{\lambda}\ln\left(1 + \frac{V_T}{\widehat{G}_{\max}^2}\right) + \frac{72\widehat{G}_{\max}^2}{\lambda} + \frac{36\|\nabla f_1(\mathbf{x}_1)\|^2}{\lambda}$$

$$+ \frac{144LL_\nu D^{1+\nu}}{\lambda}\ln\left(1 + \frac{288L}{\lambda}\right) + \frac{144L\delta(1+\ln T)}{\lambda},$$

where we define $\widehat{G}_{\max}^2 \triangleq \max_{t\in[T-1]}\sup_{\mathbf{x}\in\mathcal{X}}\|\nabla f_t(\mathbf{x}) - \nabla f_{t+1}(\mathbf{x})\|^2$, and use the property of $(L_\nu, \nu)$-Hölder smooth function $f_t$ that $\mathcal{D}_{f_t}(\mathbf{x}, \mathbf{y}) \leq L_\nu D^{1+\nu}$ [Nesterov, 2015]. Solving the trade-off: $L\delta\ln T = LL_\nu D^{1+\nu}$ with $L = \delta^{\frac{\nu-1}{1+\nu}}L_\nu^{\frac{2}{1+\nu}}$, we obtain $\delta = L_\nu D^{1+\nu}(\ln T)^{-1}$ and arrive at:

$$\text{REG}_T \leq \frac{36\widehat{G}_{\max}^2}{\lambda}\ln\left(1 + \frac{V_T}{\widehat{G}_{\max}^2}\right) + \frac{72\widehat{G}_{\max}^2}{\lambda} + \frac{36\|\nabla f_1(\mathbf{x}_1)\|^2}{\lambda}$$

$$+ \frac{144L_\nu^2 D^{2\nu}(\ln T)^{\frac{1-\nu}{1+\nu}}}{\lambda}\ln\left(1 + \frac{288L_\nu D^{\nu-1}(\ln T)^{\frac{1-\nu}{1+\nu}}}{\lambda}\right) + \frac{144L_\nu^2 D^{2\nu}(1+\ln T)^{\frac{1-\nu}{1+\nu}}}{\lambda}$$

$$= \mathcal{O}\left(\frac{\widehat{G}_{\max}^2}{\lambda}\log\left(1 + \frac{V_T}{\widehat{G}_{\max}^2}\right) + \frac{L_\nu^2 D^{2\nu}}{\lambda}(\log T)^{\frac{1-\nu}{1+\nu}} + \frac{\|\nabla f_1(\mathbf{x}_1)\|^2}{\lambda}\right),$$

where $\ln(1 + 288L_\nu(\ln T)^{(1-\nu)/(1+\nu)}/(\lambda D^{1-\nu})) = \mathcal{O}(1)$, because it only consists of the logarithm of the constant $L_\nu/(\lambda D^{1-\nu})$, and we treat the $\log\log T$ factor as a constant, following previous studies [Luo and Schapire, 2015; Zhao et al., 2024]. $\qquad\square$

### B.2 Useful Lemmas for Theorem 4

In this subsection, we provide the proofs of some useful lemmas for Theorem 4.

**Lemma 5** (Online-to-batch Conversion for Strongly Convex Functions). *Let the objective $\ell(\cdot) : \mathcal{X} \to \mathbb{R}$ be $\lambda$-strongly convex. By employing the online-to-batch conversion algorithm with online function $f_t(\mathbf{x}) \triangleq \alpha_t\langle\nabla\ell(\overline{\mathbf{x}}_t), \mathbf{x}\rangle + \frac{\lambda\alpha_t}{2}\|\mathbf{x} - \overline{\mathbf{x}}_t\|^2$, we have, for any $\mathbf{x}_\star \in \mathcal{X}$:*

$$\ell(\overline{\mathbf{x}}_T) - \ell(\mathbf{x}_\star) \leq \frac{1}{\alpha_{1:T}}\sum_{t=1}^{T}\left(f_t(\mathbf{x}_t) - f_t(\mathbf{x}_\star) - \alpha_{1:t-1}\mathcal{D}_\ell(\overline{\mathbf{x}}_{t-1}, \overline{\mathbf{x}}_t)\right), \tag{19}$$

*where $\mathcal{D}_\ell(\mathbf{x}, \mathbf{y}) \triangleq \ell(\mathbf{x}) - \ell(\mathbf{y}) - \langle\nabla\ell(\mathbf{y}), \mathbf{x} - \mathbf{y}\rangle$ is the Bregman divergence associated with $\ell$ for any $\mathbf{x}, \mathbf{y} \in \mathcal{X}$.*

*Proof.* This lemma is the variant of the stabilized online-to-batch conversion [Cutkosky, 2019] for strongly convex functions. We start from the equality:

$$\ell(\overline{\mathbf{x}}_T) - \ell(\mathbf{x}_\star) = \frac{\alpha_1\ell(\overline{\mathbf{x}}_1)}{\alpha_{1:T}} + \sum_{t=2}^{T}\frac{\alpha_{1:t}\ell(\overline{\mathbf{x}}_t) - \alpha_{1:t-1}\ell(\overline{\mathbf{x}}_{t-1})}{\alpha_{1:T}} - \ell(\mathbf{x}_\star)$$

$$= \frac{1}{\alpha_{1:T}}\sum_{t=1}^{T}\alpha_t(\ell(\overline{\mathbf{x}}_t) - \ell(\mathbf{x}_\star)) + \frac{1}{\alpha_{1:T}}\sum_{t=2}^{T}\alpha_{1:t-1}(\ell(\overline{\mathbf{x}}_t) - \ell(\overline{\mathbf{x}}_{t-1}))$$

$$\leq \frac{1}{\alpha_{1:T}}\sum_{t=1}^{T}\alpha_t\left(\langle\nabla\ell(\overline{\mathbf{x}}_t), \overline{\mathbf{x}}_t - \mathbf{x}_\star\rangle - \frac{\lambda}{2}\|\overline{\mathbf{x}}_t - \mathbf{x}_\star\|^2\right)$$

$$+ \frac{1}{\alpha_{1:T}}\sum_{t=2}^{T}\alpha_{1:t-1}\left(\langle\nabla\ell(\overline{\mathbf{x}}_t), \overline{\mathbf{x}}_t - \overline{\mathbf{x}}_{t-1}\rangle - \mathcal{D}_\ell(\overline{\mathbf{x}}_{t-1}, \overline{\mathbf{x}}_t)\right)$$

$$= \frac{1}{\alpha_{1:T}}\sum_{t=1}^{T}\alpha_t\left(\langle\nabla\ell(\overline{\mathbf{x}}_t), \overline{\mathbf{x}}_t - \mathbf{x}_\star\rangle - \frac{\lambda}{2}\|\overline{\mathbf{x}}_t - \mathbf{x}_\star\|^2\right)$$

$$+ \frac{1}{\alpha_{1:T}}\sum_{t=2}^{T}\alpha_t\langle\nabla\ell(\overline{\mathbf{x}}_t), \mathbf{x}_t - \overline{\mathbf{x}}_t\rangle - \frac{1}{\alpha_{1:T}}\sum_{t=2}^{T}\alpha_{1:t-1}\mathcal{D}_\ell(\overline{\mathbf{x}}_{t-1}, \overline{\mathbf{x}}_t)$$

$$\leq \frac{1}{\alpha_{1:T}}\sum_{t=1}^{T}\alpha_t\left(\langle\nabla\ell(\overline{\mathbf{x}}_t), \mathbf{x}_t - \mathbf{x}_\star\rangle + \frac{\lambda}{2}\|\mathbf{x}_t - \overline{\mathbf{x}}_t\|^2 - \frac{\lambda}{2}\|\overline{\mathbf{x}}_t - \mathbf{x}_\star\|^2\right) - \frac{1}{\alpha_{1:T}}\sum_{t=2}^{T}\alpha_{1:t-1}\mathcal{D}_\ell(\overline{\mathbf{x}}_{t-1}, \overline{\mathbf{x}}_t)$$

$$= \frac{1}{\alpha_{1:T}}\sum_{t=1}^{T}\left(f_t(\mathbf{x}_t) - f_t(\mathbf{x}_\star) - \alpha_{1:t-1}\mathcal{D}_\ell(\overline{\mathbf{x}}_{t-1}, \overline{\mathbf{x}}_t)\right),$$

where in the inequality we use the definition of $\lambda$-strong convexity and Bregman divergence, after which we use the property of $\alpha_{1:t-1}(\overline{\mathbf{x}}_{t-1} - \overline{\mathbf{x}}_t) = \alpha_t(\overline{\mathbf{x}}_t - \mathbf{x}_t)$ in Theorem 1 of Cutkosky [2019]. The second inequality is by directly adding the positive term $\frac{\lambda\alpha_t}{2\alpha_{1:T}}\|\mathbf{x}_t - \overline{\mathbf{x}}_t\|^2$. $\qquad\square$

**Lemma 6.** *For the settings in Theorem 4, $\beta_t$ is non-increasing with a lower bound $\bar{\beta}$. Denoting by $t_0$ the minimum iteration satisfying $\beta_{t_0} = \bar{\beta}$, otherwise let $t_0 = \tau + 1$. Algorithm 2 ensures:*

$$\ell(\overline{\mathbf{x}}_\tau) - \ell(\mathbf{x}_\star) \leq \frac{\|\nabla\ell(\mathbf{x}_1)\|^2}{\lambda\prod_{t=2}^{\tau}(1 + \beta_s)} + \sum_{t=t_0}^{\tau}\frac{2\bar{\beta}^2}{\lambda(1 + \bar{\beta})^{\tau-t+2}}\left\|\nabla\ell(\overline{\mathbf{x}}_t) - \nabla\ell(\overline{\mathbf{x}}_{t-1})\right\|^2. \tag{20}$$

*Proof.* With $f_t(\mathbf{x}) \triangleq \alpha_t \langle \nabla \ell(\overline{\mathbf{x}}_t), \mathbf{x} \rangle + \frac{\lambda \alpha_t}{2} \|\mathbf{x} - \overline{\mathbf{x}}_t\|^2$, by Lemma 5 we have:

$$\ell(\overline{\mathbf{x}}_\tau) - \ell(\mathbf{x}_\star) \le \frac{1}{\alpha_{1:\tau}} \sum_{t=1}^{\tau} \left( f_t(\mathbf{x}_t) - f_t(\mathbf{x}_\star) - \alpha_{1:t-1} \mathcal{D}_\ell(\overline{\mathbf{x}}_{t-1}, \overline{\mathbf{x}}_t) \right).$$

By Lemma 11, with the definitions $\eta_t = \frac{1}{\lambda \alpha_{1:t}}$, $M_t = \alpha_t \nabla \ell(\overline{\mathbf{x}}_{t-1}) + \lambda \alpha_t (\mathbf{x}_{t-1} - \widetilde{\mathbf{x}}_t)$, $\widetilde{\mathbf{x}}_t = \frac{1}{\alpha_{1:t}} (\sum_{s=1}^{t-1} \alpha_s \mathbf{x}_s + \alpha_t \mathbf{x}_{t-1})$, we arrive at

$$\sum_{t=1}^{\tau} (f_t(\mathbf{x}_t) - f_t(\mathbf{x}_\star)) - \sum_{t=1}^{\tau} \alpha_{1:t-1} \mathcal{D}_\ell(\overline{\mathbf{x}}_{t-1}, \overline{\mathbf{x}}_t)$$

$$\le \sum_{t=1}^{\tau} \langle \nabla f_t(\mathbf{x}_t), \mathbf{x}_t - \mathbf{x}_\star \rangle - \frac{\lambda}{2} \sum_{t=1}^{\tau} \alpha_t \|\mathbf{x}_t - \mathbf{x}_\star\|^2 - \sum_{t=1}^{\tau} \alpha_{1:t-1} \mathcal{D}_\ell(\overline{\mathbf{x}}_{t-1}, \overline{\mathbf{x}}_t)$$

$$\le \sum_{t=1}^{\tau} \eta_t \|\nabla f_t(\mathbf{x}_t) - M_t\|^2 - \sum_{t=1}^{\tau} \frac{1}{4\eta_t} \|\mathbf{x}_t - \mathbf{x}_{t+1}\|^2$$

$$- \sum_{t=1}^{\tau} \alpha_{1:t-1} \mathcal{D}_\ell(\overline{\mathbf{x}}_{t-1}, \overline{\mathbf{x}}_t) + \sum_{t=2}^{\tau} \left( \frac{1}{2\eta_t} - \frac{1}{2\eta_{t-1}} - \frac{\lambda \alpha_t}{2} \right) \|\mathbf{x}_t - \mathbf{x}_\star\|^2$$

$$= \sum_{t=2}^{\tau} \frac{\alpha_t^2}{\lambda \alpha_{1:t}} \|\nabla \ell(\overline{\mathbf{x}}_t) - \nabla \ell(\overline{\mathbf{x}}_{t-1}) + \lambda (\mathbf{x}_t - \mathbf{x}_{t-1} - \overline{\mathbf{x}}_t + \widetilde{\mathbf{x}}_t)\|^2 - \sum_{t=2}^{\tau} \frac{1}{4\eta_{t-1}} \|\mathbf{x}_t - \mathbf{x}_{t-1}\|^2$$

$$- \sum_{t=1}^{\tau} \alpha_{1:t-1} \mathcal{D}_\ell(\overline{\mathbf{x}}_{t-1}, \overline{\mathbf{x}}_t) + \frac{\alpha_1}{\lambda} \|\nabla \ell(\mathbf{x}_1)\|^2 \qquad \text{(by setting } \eta_t = \frac{1}{\lambda \alpha_{1:t}})$$

$$\le \sum_{t=2}^{\tau} \frac{2\alpha_t^2}{\lambda \alpha_{1:t}} \|\nabla \ell(\overline{\mathbf{x}}_t) - \nabla \ell(\overline{\mathbf{x}}_{t-1})\|^2 + \sum_{t=2}^{\tau} \frac{2\alpha_t^2 \lambda}{\alpha_{1:t}} \left\| \left(1 - \frac{\alpha_t}{\alpha_{1:t}}\right) (\mathbf{x}_t - \mathbf{x}_{t-1}) \right\|^2$$

$$- \sum_{t=2}^{\tau} \frac{\lambda \alpha_{1:t-1}}{4} \|\mathbf{x}_t - \mathbf{x}_{t-1}\|^2 - \sum_{t=1}^{\tau} \alpha_{1:t-1} \mathcal{D}_\ell(\overline{\mathbf{x}}_{t-1}, \overline{\mathbf{x}}_t) + \frac{\alpha_1}{\lambda} \|\nabla \ell(\mathbf{x}_1)\|^2$$

$$\le \sum_{t=2}^{\tau} \frac{2\alpha_t^2}{\lambda \alpha_{1:t}} \|\nabla \ell(\overline{\mathbf{x}}_t) - \nabla \ell(\overline{\mathbf{x}}_{t-1})\|^2 - \sum_{t=1}^{\tau} \alpha_{1:t-1} \mathcal{D}_\ell(\overline{\mathbf{x}}_{t-1}, \overline{\mathbf{x}}_t) + \frac{\alpha_1}{\lambda} \|\nabla \ell(\mathbf{x}_1)\|^2$$

$$+ \sum_{t=2}^{\tau} \left( \frac{2\alpha_t^2 \alpha_{1:t-1}^2 \lambda}{\alpha_{1:t}^3} - \frac{\lambda \alpha_{1:t-1}}{4} \right) \|\mathbf{x}_t - \mathbf{x}_{t-1}\|^2$$

$$\le \sum_{t=2}^{\tau} \frac{2\alpha_t^2}{\lambda \alpha_{1:t}} \|\nabla \ell(\overline{\mathbf{x}}_t) - \nabla \ell(\overline{\mathbf{x}}_{t-1})\|^2 - \sum_{t=1}^{\tau} \alpha_{1:t-1} \mathcal{D}_\ell(\overline{\mathbf{x}}_{t-1}, \overline{\mathbf{x}}_t) + \frac{\alpha_1}{\lambda} \|\nabla \ell(\mathbf{x}_1)\|^2,$$

where the last inequality is because $\beta_t \le 1$ and consequently

$$\frac{2\alpha_t^2 \alpha_{1:t-1}^2 \lambda}{\alpha_{1:t}^3} - \frac{\lambda \alpha_{1:t-1}}{4} = 2\alpha_{1:t-1} \lambda \left( \frac{\beta_t^2}{(1+\beta_t)^3} - \frac{1}{8} \right) \le 0.$$

Algorithm 2 ensures that for all $t \ge 2$, $\beta_t > \frac{1}{2} \sqrt{\frac{\lambda}{4L_\ell}}$, and either $\beta_t > \bar{\beta}$ or $\beta_t = \bar{\beta}$. When $\beta_t > \bar{\beta}$, it holds that $\beta_t \le \sqrt{\frac{\lambda}{4L_t}}$ due to the algorithm design, then we have

$$\frac{2\alpha_t^2}{\lambda \alpha_{1:t}} \|\nabla \ell(\overline{\mathbf{x}}_t) - \nabla \ell(\overline{\mathbf{x}}_{t-1})\|^2 - \alpha_{1:t-1} \mathcal{D}_\ell(\overline{\mathbf{x}}_{t-1}, \overline{\mathbf{x}}_t)$$

$$= \left( \frac{4L_t \alpha_t^2}{\lambda \alpha_{1:t} \alpha_{1:t-1}} - 1 \right) \alpha_{1:t-1} \mathcal{D}_\ell(\overline{\mathbf{x}}_{t-1}, \overline{\mathbf{x}}_t) = \left( \frac{4L_t \beta_t^2}{\lambda (1 + \beta_t)} - 1 \right) \alpha_{1:t-1} \mathcal{D}_\ell(\overline{\mathbf{x}}_{t-1}, \overline{\mathbf{x}}_t) \le 0.$$

Since $\beta_t$ is non-increasing, denoting by $t_0$ the minimum iteration satisfying $\beta_{t_0} = \bar{\beta}$, otherwise let $t_0 = \tau + 1$. Then for all $t \ge t_0$, $\beta_t = \bar{\beta}$. Finally, we arrive at

$$\ell(\overline{\mathbf{x}}_\tau) - \ell(\mathbf{x}_\star) \le \frac{\alpha_1 \|\nabla \ell(\mathbf{x}_1)\|^2}{\lambda \alpha_{1:\tau}} + \sum_{t=t_0}^{\tau} \frac{2\alpha_t^2}{\lambda \alpha_{1:t} \alpha_{1:\tau}} \|\nabla \ell(\overline{\mathbf{x}}_t) - \nabla \ell(\overline{\mathbf{x}}_{t-1})\|^2$$

$$= \frac{\|\nabla\ell(\mathbf{x}_1)\|^2}{\lambda\prod_{t=2}^{\tau}(1+\beta_s)} + \sum_{t=t_0}^{\tau}\frac{2\bar{\beta}^2}{\lambda(1+\bar{\beta})^{\tau-t+2}}\left\|\nabla\ell(\overline{\mathbf{x}}_t) - \nabla\ell(\overline{\mathbf{x}}_{t-1})\right\|^2,$$

which finishes the proof. $\qquad\square$

## B.3 Proof of Theorem 4 and Corollary

*Proof of Theorem 4.* We do not know whether $\ell(\mathbf{x})$ is smooth or non-smooth, but it is Lipschitz continuous with unknown constant $G$. We have $\max_{1 < t \leq \tau}\|\nabla\ell(\overline{\mathbf{x}}_t) - \nabla\ell(\overline{\mathbf{x}}_{t-1})\|^2 \leq 4G^2$. By Lemma 6,

$$\ell(\overline{\mathbf{x}}_\tau) - \ell(\mathbf{x}_\star) \leq \frac{\|\nabla\ell(\mathbf{x}_1)\|^2}{\lambda\prod_{t=2}^{\tau}(1+\beta_s)} + \frac{8G^2\bar{\beta}^2}{\lambda}\sum_{t=t_0}^{\tau}\frac{1}{(1+\bar{\beta})^{\tau-t+2}}$$

$$\leq \frac{2\|\nabla\ell(\mathbf{x}_1)\|^2}{\lambda(1+\max\{1/(4\sqrt{\kappa}),\bar{\beta}\})^\tau} + \frac{8G^2\bar{\beta}}{\lambda}\cdot\mathbb{1}\{t_0 \leq \tau\}. \tag{21}$$

By choosing $\bar{\beta} = \exp(\frac{1}{T}\ln T) - 1$, we conduct the following case-by-case study:

**Case of $\frac{1}{4\sqrt{\kappa}} \geq \bar{\beta}$.** Then since for all $t \geq 2$, $\beta_t > \frac{1}{4\sqrt{\kappa}}$, we have $t_0 = \tau + 1$ by definition, then the second term in (21) becomes zero. In this case, we have

$$\ell(\overline{\mathbf{x}}_\tau) - \ell(\mathbf{x}_\star) \leq \mathcal{O}\left(\frac{\|\nabla\ell(\mathbf{x}_1)\|^2}{\lambda}\min\left\{\exp\left(\frac{-\tau}{1+4\sqrt{\kappa}}\right), (1+\bar{\beta})^{-\tau}\right\}\right).$$

Moreover, the total gradient queries number $T \leq \tau + \lfloor\log_2(4\sqrt{\kappa})\rfloor$, then we arrive at

$$\ell(\overline{\mathbf{x}}_\tau) - \ell(\mathbf{x}_\star) \leq \mathcal{O}\left(\frac{\|\nabla\ell(\mathbf{x}_1)\|^2}{\lambda}\min\left\{\exp\left(\frac{-T+\log_2(4\sqrt{\kappa})}{1+4\sqrt{\kappa}}\right), (1+\bar{\beta})^{-T+\log_2(4\sqrt{\kappa})}\right\}\right)$$

$$\leq \mathcal{O}\left(\frac{\|\nabla\ell(\mathbf{x}_1)\|^2}{\lambda}\min\left\{\exp\left(\frac{-T}{1+4\sqrt{\kappa}}\right), \frac{1}{T}(1+\bar{\beta})^{\log_2(1/\bar{\beta})}\right\}\right)$$

$$\leq \mathcal{O}\left(\frac{\|\nabla\ell(\mathbf{x}_1)\|^2}{\lambda}\min\left\{\exp\left(\frac{-T}{6\sqrt{\kappa}}\right), \frac{1}{T}\right\}\right),$$

where in the last inequality we use $(1+x)^{1+\log_2(1/x)} < 3$ for all $x > 0$.

**Case of $\frac{1}{4\sqrt{\kappa}} < \bar{\beta}$.** In this case, by (21) we have:

$$\ell(\overline{\mathbf{x}}_\tau) - \ell(\mathbf{x}_\star) \leq \mathcal{O}\left(\frac{\|\nabla\ell(\mathbf{x}_1)\|^2}{\lambda}(1+\bar{\beta})^{-\tau} + \frac{G^2\bar{\beta}}{\lambda}\right).$$

Moreover, the total gradient queries number $T \leq \tau + \lceil\log_2(1/\bar{\beta})\rceil$, then we arrive at

$$\ell(\overline{\mathbf{x}}_T) - \ell(\mathbf{x}_\star) \leq \mathcal{O}\left(\frac{\|\nabla\ell(\mathbf{x}_1)\|^2}{\lambda}(1+\bar{\beta})^{-T+\lceil\log_2(1/\bar{\beta})\rceil} + \frac{G^2\bar{\beta}}{\lambda}\right)$$

$$\leq \mathcal{O}\left(\frac{\|\nabla\ell(\mathbf{x}_1)\|^2}{\lambda T} + \frac{G^2\log T}{\lambda T}\right),$$

where we use $(1+x)^{1+\log_2(1/x)} < 3$ for all $x > 0$ and $\bar{\beta} = \exp(\frac{1}{T}\ln T) - 1 \leq \frac{5}{4T}\ln T$. Additionally, the exponential rate, that is $\exp(\frac{-T}{6\sqrt{\kappa}}) > \exp(-\frac{4}{6}T\bar{\beta}) \geq \exp(-\frac{5}{6}\ln T) = \frac{1}{T^{5/6}} = \Omega(\frac{\log T}{T})$, is dominated. Finally, combining these two cases, we obtain

$$\ell(\overline{\mathbf{x}}_\tau) - \ell(\mathbf{x}_\star) \leq \mathcal{O}\left(\frac{G^2}{\lambda}\min\left\{\exp\left(\frac{-T}{6\sqrt{\kappa}}\right), \frac{\log T}{T}\right\}\right),$$

which finishes the proof. $\qquad\square$

When the optimization problem is easier, i.e., with additional informations, we can use Algorithm 2 framework to obtain better convergence rates, as provided in Corollary 1.

**Corollary 1.** *Consider the optimization problem $\min_{\mathbf{x} \in \mathcal{X}} \ell(\mathbf{x})$ in the deterministic setting, where the objective $\ell$ is $\lambda$-strongly convex. In the following two cases:*

*(i) If $\ell$ is known to be $L_\ell$-smooth, then [Algorithm 2](#) with $\beta_1 = \bar{\beta} = \sqrt{\lambda/(4L_\ell)}$ ensures that*

$$\ell(\overline{\mathbf{x}}_\tau) - \ell(\mathbf{x}_\star) \leq \mathcal{O}\left(\frac{\|\nabla\ell(\mathbf{x}_1)\|^2}{\lambda}\exp\left(\frac{-T}{1+2\sqrt{\kappa}}\right)\right),$$

*where $\kappa \triangleq L_\ell/\lambda$ denotes the condition number.*

*(ii) If $\ell$ is smooth but the smoothness parameter $L_\ell$ remains unknown, then [Algorithm 2](#) with $\beta_1 = 1, \bar{\beta} = 0$ ensures that*

$$\ell(\overline{\mathbf{x}}_\tau) - \ell(\mathbf{x}_\star) \leq \mathcal{O}\left(\frac{\|\nabla\ell(\mathbf{x}_1)\|^2}{\lambda}\exp\left(\frac{-T}{1+4\sqrt{\kappa}}\right)\right).$$

Interestingly, in the first case of [Corollary 1](#), where $L_\ell$ is known, our convergence rate matches [Wei and Chen](#) [2025, Theorem 1.1]. Moreover, their "over-relaxation" update form coincides with the *one-step* update variant of our optimistic OGD online algorithm.

*Proof.* **The first case.** We are given the smoothness parameter $L_\ell$. By [Lemma 6](#), since $\beta_t \equiv \sqrt{\lambda/(4L_\ell)} \leq \sqrt{\lambda/(4L_t)}$ for all $t \geq 2$, we have $t_0 = \tau + 1$ by definition, and $\tau = T$, therefore

$$\ell(\overline{\mathbf{x}}_T) - \ell(\mathbf{x}_\star) \leq \frac{2\|\nabla\ell(\mathbf{x}_1)\|^2}{\lambda\big(1 + \sqrt{\lambda/(4L_\ell)}\big)^T} \leq \mathcal{O}\left(\frac{\|\nabla\ell(\mathbf{x}_1)\|^2}{\lambda}\exp\left(\frac{-T}{1+2\sqrt{\kappa}}\right)\right),$$

where we use $(1 + x^{-1})^{-T} = (1 - 1/(1+x))^T \leq \exp(-T/(1+x))$ for all $x > 0$.

**The second case.** We know that $\ell$ is smooth but do not know the exact smoothness parameter $L_\ell$. With $\bar{\beta} = 0$, we have $\frac{1}{4\sqrt{\kappa}} \leq \beta_t \leq \sqrt{\frac{\lambda}{4L_t}}$ for all $2 \leq t \leq \tau$. By [Lemma 6](#) with $t_0 = \tau + 1$,

$$\ell(\overline{\mathbf{x}}_\tau) - \ell(\mathbf{x}_\star) \leq \frac{\|\nabla\ell(\mathbf{x}_1)\|^2}{\lambda\prod_{t=2}^\tau(1+\beta_s)} \leq \frac{2\|\nabla\ell(\mathbf{x}_1)\|^2}{\lambda(1+1/(4\sqrt{\kappa}))^\tau} \leq \frac{2\|\nabla\ell(\mathbf{x}_1)\|^2}{\lambda}\exp\left(\frac{-\tau}{1+4\sqrt{\kappa}}\right),$$

where we use $(1 + x^{-1})^{-\tau} = (1 - 1/(1+x))^\tau \leq \exp(-\tau/(1+x))$ for all $x > 0$. Moreover, the total gradient queries number $T \leq \tau + \lfloor\log_2(4\sqrt{\kappa})\rfloor$, substituting into the above inequality,

$$\ell(\overline{\mathbf{x}}_\tau) - \ell(\mathbf{x}_\star) \leq \frac{2\|\nabla\ell(\mathbf{x}_1)\|^2}{\lambda}\exp\left(\frac{-T}{1+4\sqrt{\kappa}}\right)\exp\left(\frac{\log_2(4\sqrt{\kappa})}{1+4\sqrt{\kappa}}\right)$$
$$< \frac{3\|\nabla\ell(\mathbf{x}_1)\|^2}{\lambda}\exp\left(\frac{-T}{1+4\sqrt{\kappa}}\right),$$

where we use $\exp\left(\frac{\log_2 x}{1+x}\right) < 1.5$ for all $x > 0$. This case is proved. $\square$

### B.4 Proof of Theorem 5 and Discussions

*Proof of [Theorem 5](#).* For any $\mathbf{x} \in \mathcal{X}$, we have $\ell(\mathbf{x}) - \ell(\mathbf{x}_\star) \leq \|\nabla\ell(\mathbf{x})\|\|\mathbf{x} - \mathbf{x}_\star\| \leq \frac{1}{\lambda}\|\nabla\ell(\mathbf{x})\|^2$ because $\ell(\mathbf{x})$ is $\lambda$-strongly convex, and $\nabla\ell(\mathbf{x}_\star) = \mathbf{0}$. Hence when $\kappa > T^2$, the convergence rate of $\frac{1}{\lambda}\|\nabla\ell(\mathbf{x})\|^2\exp(\frac{-T}{\sqrt{\kappa}}) \geq \frac{1}{\lambda e}\|\nabla\ell(\mathbf{x})\|^2$ becomes vacuous. Therefore, without loss of generality, we assume $\kappa < T^2$.

Moreover, by calculating the curvature estimate $\widehat{\lambda} = \frac{\|\nabla\ell(\mathbf{a}) - \nabla\ell(\mathbf{b})\|}{\|\mathbf{a}-\mathbf{b}\|}$ with any $\mathbf{a}, \mathbf{b} \in \mathbb{R}^d$, we have $\lambda \leq \widehat{\lambda} \leq L_\ell$. Combining with $\kappa < T^2$ implies that $\lambda \in \left[\widehat{\lambda}/T^2, \widehat{\lambda}\right]$.

Denoting by $M = \lceil 2\log_2 T\rceil$, and $\lambda_i = 2^{-i} \cdot \widehat{\lambda}$ for $i \in [M]$, there exists $i_\star \in [M]$ that $\lambda_{i_\star} \leq \lambda \leq 2\lambda_{i_\star}$. Then $\ell(\cdot)$ is also $\lambda_{i_\star}$-strongly convex with condition number being $2\kappa$. Substituting into [Theorem 4](#) with $T_i = \frac{T}{M}$, we have

$$\ell(\overline{\mathbf{x}}^{i_\star}) - \ell(\mathbf{x}_\star) \leq \mathcal{O}\left(\frac{\|\nabla\ell(\mathbf{x}_1)\|^2}{\lambda}\exp\left(\frac{-T}{(1+4\sqrt{2\kappa})\lceil 2\log_2 T\rceil}\right)\right).$$

The proof is finished. $\square$

**Comparison with Lan et al. [2023]**   We compare our result in Theorem 5 with the sample complexity bound for optimizing the gradient norm established in Theorem 5.1 of Lan et al. [2023], that is, with $C_1 = \sqrt{2}(3 + 16\sqrt{2c_\mathcal{A}}), c_\mathcal{A} = 4$ as they provided, $T \leq (4 + 8\sqrt{5}C_1)\sqrt{\kappa}\log_2(\|\nabla\ell(\mathbf{x}_1)\|/\varepsilon) + \mathcal{O}(1)$.

First, we reformulate their result as follows:

*(i)* After translating their result into the convergence rate of the gradient norm, it turns out to be $\mathcal{O}\big(\exp(\frac{-T \cdot (\ln 2)}{(4 + 8\sqrt{5}C_1)\sqrt{\kappa}})\big)$. Substituting the constants implies:

$$\|\nabla\ell(\mathbf{x}_T)\| \leq \mathcal{O}\left(\exp\left(\frac{-T}{1766\sqrt{\kappa}}\right)\right).$$

*(ii)* Applying $\ell(\mathbf{x}_T) - \ell(\mathbf{x}_\star) \leq \frac{1}{\lambda}\|\nabla\ell(\mathbf{x}_T)\|^2$, we obtain a sub-optimality bound given by

$$\ell(\mathbf{x}_T) - \ell(\mathbf{x}_\star) \leq \mathcal{O}\left(\exp\left(\frac{-T}{882\sqrt{\kappa}}\right)\right). \tag{22}$$

Then we consider when our rate in Theorem 5 is better than Eq. (22). Solving the following condition:

$$(1 + 4\sqrt{2\kappa})\lceil 2\log_2 T\rceil \leq (1 + 4\sqrt{2})\lceil 2\log_2 T\rceil\sqrt{\kappa} \leq 882\sqrt{\kappa},$$

implies that $T \leq 8.7 \times 10^{19}$.

## C   Supporting Lemmas

In this section, we provide supporting lemmas for this paper.

### C.1   Lemmas for Optimistic OGD Algorithms

In this part, we provide useful lemmas for optimistic OGD and its one-step variant.

**Lemma 7** (Proposition 7 of Chiang et al. [2012]). *Consider the following two updates: (i)* $\mathbf{x} = \arg\min_{\mathbf{x}\in\mathcal{X}} \{\langle\mathbf{g}, \mathbf{x}\rangle + \mathcal{D}_\psi(\mathbf{x}, \mathbf{c})\}$, *and (ii)* $\mathbf{x}' = \arg\min_{\mathbf{x}\in\mathcal{X}} \{\langle\mathbf{g}', \mathbf{x}\rangle + \mathcal{D}_\psi(\mathbf{x}, \mathbf{c})\}$, *where the regularizer* $\psi : \mathcal{X} \to \mathbb{R}$ *is* $\lambda$-*strongly convex function with respect to* $\|\cdot\|$, *we have* $\lambda\|\mathbf{x} - \mathbf{x}'\| \leq \|\mathbf{g} - \mathbf{g}'\|_*$.

**Lemma 8** (Bregman proximal inequality, Lemma 3.2 of Chen and Teboulle [1993]). *Consider the following update:* $\mathbf{x} = \arg\min_{\mathbf{x}\in\mathcal{X}} \{\langle\mathbf{g}, \mathbf{x}\rangle + \mathcal{D}_\psi(\mathbf{x}, \mathbf{c})\}$, *where the regularizer* $\psi : \mathcal{X} \to \mathbb{R}$ *is convex function, then for all* $\mathbf{u} \in \mathcal{X}$, *we have* $\langle\mathbf{g}, \mathbf{x} - \mathbf{u}\rangle \leq \mathcal{D}_\psi(\mathbf{u}, \mathbf{c}) - \mathcal{D}_\psi(\mathbf{u}, \mathbf{x}) - \mathcal{D}_\psi(\mathbf{x}, \mathbf{c})$.

**Lemma 9** (Theorem 1 of Zhao et al. [2024]). *Under Assumption 1, Optimistic OGD specialized at Eq. (6), that starts at* $\widehat{\mathbf{x}}_1 \in \mathcal{X}$ *and updates by*

$$\mathbf{x}_t = \Pi_\mathcal{X}\left[\widehat{\mathbf{x}}_t - \eta_t M_t\right], \quad \widehat{\mathbf{x}}_{t+1} = \Pi_\mathcal{X}\left[\widehat{\mathbf{x}}_t - \eta_t \nabla f_t(\mathbf{x}_t)\right],$$

*ensures that*

$$\sum_{t=1}^{T}\langle\nabla f_t(\mathbf{x}_t), \mathbf{x}_t - \mathbf{u}_t\rangle \leq \underbrace{\sum_{t=1}^{T}\langle\nabla f_t(\mathbf{x}_t) - M_t, \mathbf{x}_t - \widehat{\mathbf{x}}_{t+1}\rangle}_{\text{TERM-A}} + \underbrace{\sum_{t=1}^{T}\frac{1}{2\eta_t}\left(\|\mathbf{u}_t - \widehat{\mathbf{x}}_t\|^2 - \|\mathbf{u}_t - \widehat{\mathbf{x}}_{t+1}\|^2\right)}_{\text{TERM-B}}$$

$$- \underbrace{\sum_{t=1}^{T}\frac{1}{2\eta_t}\left(\|\mathbf{x}_t - \widehat{\mathbf{x}}_{t+1}\|^2 + \|\mathbf{x}_t - \widehat{\mathbf{x}}_t\|^2\right)}_{\text{TERM-C}}, \tag{23}$$

*where* $\mathbf{u}_1, \ldots, \mathbf{u}_T \in \mathcal{X}$ *are arbitrary comparators.*

**Lemma 10.** *Under Assumption 1, Optimistic OGD specialized at Eq. (6) with non-increasing step sizes* $\eta_t$, *ensures that*

$$\sum_{t=1}^{T}\langle\nabla f_t(\mathbf{x}_t), \mathbf{x}_t - \mathbf{u}_t\rangle \leq \sum_{t=1}^{T}\eta_{t+1}\|\nabla f_t(\mathbf{x}_t) - M_t\|^2 + \frac{D^2 + DP_T}{\eta_{T+1}} - \sum_{t=2}^{T}\frac{1}{8\eta_{t+1}}\|\mathbf{x}_t - \mathbf{x}_{t-1}\|^2, \tag{24}$$

*where* $P_T \triangleq \sum_{t=2}^{T}\|\mathbf{u}_t - \mathbf{u}_{t-1}\|$ *is the path length.*

**Lemma 11** (One-step Variant of Optimistic OGD, [Joulani et al., 2020a])**.** *Under Assumption 1, the one-step variant of optimistic OGD that starts at $\mathbf{x}_1 \in \mathcal{X}$ and updates by*

$$\mathbf{x}_{t+1} = \Pi_{\mathcal{X}} \left[ \mathbf{x}_t - \eta_t (\nabla f_t(\mathbf{x}_t) - M_t + M_{t+1}) \right], \tag{25}$$

*ensures that, for all $\mathbf{u} \in \mathcal{X}$:*

$$\sum_{t=1}^{T} \langle \nabla f_t(\mathbf{x}_t), \mathbf{x}_t - \mathbf{u} \rangle \leq \sum_{t=1}^{T} \left( \langle \nabla f_t(\mathbf{x}_t) - M_t, \mathbf{x}_t - \mathbf{x}_{t+1} \rangle - \frac{1}{2\eta_t} \|\mathbf{x}_t - \mathbf{x}_{t+1}\|^2 \right)$$
$$+ \sum_{t=2}^{T} \left( \frac{1}{2\eta_t} - \frac{1}{2\eta_{t-1}} \right) \|\mathbf{x}_t - \mathbf{u}\|^2 + \frac{1}{2\eta_1} \|\mathbf{x}_1 - \mathbf{u}\|^2.$$

*Proof of Lemma 10.* By Lemma 9, we consider each term:

$$\text{TERM-A} \leq \sum_{t=1}^{T} \eta_{t+1} \|\nabla f_t(\mathbf{x}_t) - M_t\|_*^2 + \sum_{t=1}^{T} \left( \frac{1}{4\eta_{t+1}} - \frac{1}{4\eta_t} \right) \|\mathbf{x}_t - \widehat{\mathbf{x}}_{t+1}\|^2 + \sum_{t=1}^{T} \frac{1}{4\eta_t} \|\mathbf{x}_t - \widehat{\mathbf{x}}_{t+1}\|^2$$

$$\leq \sum_{t=1}^{T} \eta_{t+1} \|\nabla f_t(\mathbf{x}_t) - M_t\|_*^2 + \frac{D^2}{4\eta_{T+1}} + \sum_{t=1}^{T} \frac{1}{4\eta_t} \|\mathbf{x}_t - \widehat{\mathbf{x}}_{t+1}\|^2,$$

$$\text{TERM-B} \leq \frac{D^2}{2\eta_1} + \sum_{t=2}^{T} \left( \frac{1}{2\eta_t} \|\mathbf{u}_t - \widehat{\mathbf{x}}_t\|^2 - \frac{1}{2\eta_t} \|\mathbf{u}_{t-1} - \widehat{\mathbf{x}}_t\|^2 + \frac{1}{2\eta_t} \|\mathbf{u}_{t-1} - \widehat{\mathbf{x}}_t\|^2 - \frac{1}{2\eta_{t-1}} \|\mathbf{u}_{t-1} - \widehat{\mathbf{x}}_t\|^2 \right)$$

$$\leq \frac{D^2}{2\eta_1} + \sum_{t=2}^{T} \frac{1}{2\eta_t} \left( \|\mathbf{u}_t - \widehat{\mathbf{x}}_t\|^2 - \|\mathbf{u}_{t-1} - \widehat{\mathbf{x}}_t\|^2 \right) + \sum_{t=2}^{T} \left( \frac{1}{2\eta_t} - \frac{1}{2\eta_{t-1}} \right) D^2$$

$$\leq \frac{D^2}{2\eta_T} + \sum_{t=2}^{T} \frac{1}{2\eta_t} \|\mathbf{u}_t - \mathbf{u}_{t-1}\| \cdot \|\mathbf{u}_t - \widehat{\mathbf{x}}_t + \mathbf{u}_{t-1} - \widehat{\mathbf{x}}_t\|$$

$$\leq \frac{D^2}{2\eta_{T+1}} + \frac{D P_T}{\eta_{T+1}},$$

$$\text{TERM-C} \geq \sum_{t=1}^{T} \frac{1}{4\eta_t} \|\mathbf{x}_t - \widehat{\mathbf{x}}_{t+1}\|^2 + \sum_{t=2}^{T} \frac{1}{4\eta_{t-1}} \left( \|\mathbf{x}_{t-1} - \widehat{\mathbf{x}}_t\|^2 + \|\mathbf{x}_t - \widehat{\mathbf{x}}_t\|^2 \right)$$

$$\geq \sum_{t=1}^{T} \frac{1}{4\eta_t} \|\mathbf{x}_t - \widehat{\mathbf{x}}_{t+1}\|^2 + \sum_{t=2}^{T} \left( \frac{1}{8\eta_{t-1}} - \frac{1}{8\eta_t} \right) \|\mathbf{x}_t - \mathbf{x}_{t-1}\|^2 + \sum_{t=2}^{T} \frac{1}{8\eta_t} \|\mathbf{x}_t - \mathbf{x}_{t-1}\|^2$$

$$\geq \sum_{t=1}^{T} \frac{1}{4\eta_t} \|\mathbf{x}_t - \widehat{\mathbf{x}}_{t+1}\|^2 - \frac{D^2}{8\eta_T} + \sum_{t=2}^{T} \left( \frac{1}{8\eta_t} - \frac{1}{8\eta_{t+1}} \right) \|\mathbf{x}_t - \mathbf{x}_{t-1}\|^2 + \sum_{t=2}^{T} \frac{1}{8\eta_{t+1}} \|\mathbf{x}_t - \mathbf{x}_{t-1}\|^2$$

$$\geq \sum_{t=1}^{T} \frac{1}{4\eta_t} \|\mathbf{x}_t - \widehat{\mathbf{x}}_{t+1}\|^2 - \frac{D^2}{4\eta_{T+1}} + \sum_{t=2}^{T} \frac{1}{8\eta_{t+1}} \|\mathbf{x}_t - \mathbf{x}_{t-1}\|^2,$$

where we apply Assumption 1 and the condition that $\eta_t$ is non-increasing. Combining TERM-A, TERM-B and TERM-C finishes the proof. $\square$

*Proof of Lemma 11.* By Lemma 8 with $\psi(\mathbf{x}) = \frac{1}{2\eta} \|\mathbf{x}\|^2$, the update Eq. (25) implies for all $\mathbf{u} \in \mathcal{X}$:

$$\langle \nabla f_t(\mathbf{x}_t) - M_t + M_{t+1}, \mathbf{x}_{t+1} - \mathbf{u} \rangle \leq \frac{1}{2\eta_t} \left( \|\mathbf{u} - \mathbf{x}_t\|^2 - \|\mathbf{u} - \mathbf{x}_{t+1}\|^2 - \|\mathbf{x}_t - \mathbf{x}_{t+1}\|^2 \right).$$

Then by rearranging and taking summation from $t = 1$ to $T$, we arrive at:

$$\sum_{t=1}^{T} \langle \nabla f_t(\mathbf{x}_t), \mathbf{x}_t - \mathbf{u} \rangle$$

$$\leq \sum_{t=1}^{T} \langle \nabla f_t(\mathbf{x}_t) - M_t, \mathbf{x}_t - \mathbf{x}_{t+1} \rangle + \langle M_1, \mathbf{x}_1 \rangle - \langle M_{T+1}, \mathbf{x}_{T+1} \rangle + \sum_{t=1}^{T} \langle M_{t+1} - M_t, \mathbf{u} \rangle$$

$$+ \sum_{t=2}^{T} \left( \frac{1}{2\eta_t} - \frac{1}{2\eta_{t-1}} \right) \|\mathbf{x}_t - \mathbf{u}\|^2 + \frac{1}{2\eta_1} \|\mathbf{x}_1 - \mathbf{u}\|^2 - \sum_{t=1}^{T} \frac{1}{2\eta_t} \|\mathbf{x}_t - \mathbf{x}_{t+1}\|^2$$

$$\leq \sum_{t=1}^{T} \left( \langle \nabla f_t(\mathbf{x}_t) - M_t, \mathbf{x}_t - \mathbf{x}_{t+1} \rangle - \frac{1}{2\eta_t} \|\mathbf{x}_t - \mathbf{x}_{t+1}\|^2 \right)$$

$$+ \sum_{t=2}^{T} \left( \frac{1}{2\eta_t} - \frac{1}{2\eta_{t-1}} \right) \|\mathbf{x}_t - \mathbf{u}\|^2 + \frac{1}{2\eta_1} \|\mathbf{x}_1 - \mathbf{u}\|^2,$$

where we define $M_1 \triangleq \mathbf{0}$ and $M_{T+1} \triangleq \mathbf{0}$. $\qquad\square$

## C.2 Useful Lemmas

This part provides some useful lemmas for mathematical analysis.

**Lemma 12** (McMahan and Streeter [2010]). *Suppose non-negative sequence $\{a_t\}_{t=1}^{T}$ and constant $\delta > 0$, then we have*

$$\sum_{t=1}^{T} \frac{a_t}{\sqrt{\delta + \sum_{s=1}^{t} a_s}} \leq 2\sqrt{\delta + \sum_{t=1}^{T} a_t}. \qquad (26)$$

**Lemma 13.** *Suppose non-negative sequence $\{a_t\}_{t=1}^{T}$. Define $a_{\max} = \max_{t \in [T]} a_t$ and assume $a_{\max} > 0$, then we have*

$$\sum_{t=1}^{T} \frac{a_t}{t} \leq a_{\max} \ln \left( 1 + \frac{1}{a_{\max}} \sum_{t=1}^{T} a_t \right) + 2a_{\max}.$$

**Lemma 14.** *Suppose $A > 0$ and non-negative sequence $\{b_t\}_{t=1}^{T}$ and denote by $b_{\max} = \max_{t \in [T]} b_t > 0$. Then it holds that*

$$\sum_{t=1}^{T} \left( \frac{A}{t} - 1 \right) b_t \leq b_{\max} \cdot A \ln(1 + A).$$

*Proof of Lemma 13.* Define $\tau = \lceil \frac{1}{a_{\max}} \sum_{t=1}^{T} a_t \rceil \in [T]$. We have

$$\sum_{t=1}^{\tau} \frac{a_t}{t} \leq a_{\max} \sum_{t=1}^{\tau} \frac{1}{t} \leq a_{\max} \left( 1 + \int_{x=1}^{\tau} \frac{1}{x} \mathrm{d}x \right) \leq a_{\max} \ln \left( 1 + \frac{1}{a_{\max}} \sum_{t=1}^{T} a_t \right) + a_{\max}.$$

If $\tau < T$, we also have

$$\sum_{t=\tau+1}^{T} \frac{a_t}{t} \leq \frac{1}{\tau} \sum_{t=\tau+1}^{T} a_t \leq \frac{a_{\max}}{\sum_{t=1}^{T} a_t} \sum_{t=\tau+1}^{T} a_t = a_{\max}.$$

$\qquad\square$

*Proof of Lemma 14.* Define $\tau = \min\{T, \lfloor A \rfloor\}$ and trivially assume $\tau \geq 1$, then we have

$$\frac{1}{b_{\max}} \sum_{t=1}^{T} \left( \frac{A}{t} - 1 \right) b_t \leq \sum_{t=1}^{\tau} \left( \frac{A}{t} - 1 \right) \leq A \left( 1 + \int_{s=1}^{\tau} \frac{1}{s} \mathrm{d}s \right) - \tau = A + A \ln \tau - \tau,$$

whose maximum is $A \ln A \leq A \ln(1 + A)$. $\qquad\square$

