# OpenReview forum: "Gradient-Variation Online Adaptivity for Accelerated Optimization with Hölder Smoothness"
_NeurIPS.cc/2025/Conference — NeurIPS 2025 spotlight_

### Official Review · Reviewer_BMFE · 2025-06-26

**Clarity:** 3
**Significance:** 2
**Originality:** 2
**Rating:** 5
**Confidence:** 4

**Summary:**

This paper studies regret bounds with gradient variation in the context of online learning.

It first extends existing regret bounds by incorporating gradient variation terms when the loss functions are Hölder smooth and Lipschitz continuous. The authors then provide an online-to-batch conversion technique that yields accelerated convergence rates of order $O(T^{-2})$ for smooth functions, thereby generalizing existing results. A key strength of the method is its universality—it does not require prior knowledge of the Lipschitz or smoothness constants.

The authors further extend their results to the strongly convex setting, deriving regret bounds and corresponding accelerated online-to-batch conversion rates. However, in this case, the method is not fully universal, as it requires prior knowledge of the strong convexity parameter. Additionally, the obtained rates are slightly suboptimal when the Lipschitz constant is unknown.

Finally, the authors design a meta-algorithm that achieves dynamic regret bounds with gradient variation when the loss functions are Lipschitz and Hölder smooth. They also propose an algorithm that adapts to unknown strong convexity parameter.

**Questions:**

1. Could the authors elaborate on the specific challenges involved in extending the online-to-batch conversion to the stochastic setting in the strongly convex case? Are there particular obstacles that prevent a direct extension of the techniques used in the smooth or convex cases?

2. Is it possible to obtain an online-to-batch conversion for Theorem 6 in the case where the curvature parameter (i.e., the strong convexity constant) is unknown? If not, could the authors comment on what makes this technically difficult or whether a universal algorithm could adapt to unknown curvature in this setting?

**Ethical Concerns:**

["NO or VERY MINOR ethics concerns only"]

**Final Justification:**

I read other reviews and rebuttal. The authors address well my concerns and no reviewer seem to raise major weakness. I lean toward acceptance and raise my score to 5.

**Limitations:**

Yes

**Quality:**

3

**Strengths And Weaknesses:**

Strengths:

The paper is well written and presents a clearly motivated framework, relevant to both online learning and convex optimization. It offers a substantial analysis of regret bounds incorporating gradient variation for Hölder smooth functions, providing a range of results that either improve upon existing work (by requiring fewer parameter assumptions) or generalize it (by considering Hölder smoothness with a generic parameter $\nu$). These contributions add breadth and flexibility to the theory of learning with smooth losses.

Weaknesses:

It is not entirely clear to what extent the results go beyond incremental generalizations of existing work—primarily extending known analyses from $\nu = 1$ (standard smoothness) to a generic Hölder smoothness parameter $\nu$. Could the authors clarify what technical challenges were encountered and how they were addressed in order to extend the results? For example, the analysis in Theorem 1 appears to follow standard Optimistic OMD techniques, and the online-to-batch conversion arguments seem to closely mirror those in [Cutkosky, 2019].

---

> ### Author Rebuttal · Authors · 2025-07-30
>
> Thank you for the valuable feedback. Below, we will respond to your questions and clarify our contributions in hopes of addressing your concerns.
>
> ---
>
> **Q1.** It is not entirely clear to what extent the results go beyond incremental generalizations of existing work—primarily extending known analyses from $\nu=1$ (standard smoothness) to a generic Hölder smoothness parameter $\nu$. Could the authors clarify what technical challenges were encountered and how they were addressed in order to extend the results? For example, the analysis in Theorem 1 appears to follow standard Optimistic OMD techniques, and the online-to-batch conversion arguments seem to closely mirror those in [Cutkosky, 2019].
>
> **A1.** Thanks for the comment. Below, we will explain the contributions of ours. There are some noteworthy points, such as Theorem 4.
>
> **Results:**
>
> - **About Theorem 1,3,5,6:** These regret bounds interpolated through the Hölder smoothness parameters $(L_\nu, \nu)$ are novel and meaningful. They bridge the classic gradient variation bound and worst-case bound, and for the first time, demonstrate a connection between these two types of bounds across various settings (convex, strongly convex, dynamic regret, and universal).
> - **About Theorem 4:** It is particularly noteworthy. It is very challenging to develop an algorithm that can adapt to the smoothness level to achieve accelerated convergence while simultaneously guaranteeing nearly optimal convergence in the non-smooth case. To the best of our knowledge, the most recent results date back to the 2017 paper [1], which, however, only obtained a *non-accelerated* $O(\exp(-T/\kappa)\cdot T/\kappa)$ rate for the smooth case. We improved this to an *accelerated and near-optimal* $O(\exp(-T/(\sqrt{\kappa}\cdot\log \kappa)))$ rate.
>
> **Techniques:**
>
> - We applied the adaptive step size and a case-by-case analysis technique [2] to gradient variation online learning (where previous works often required the algorithm to explicitly know the smoothness parameter $L$ for clipping the step size), and further extending it to the Hölder smooth setting. We also avoid the explicit requirement for smoothness $L$ in optimizing dynamic regret that was present in previous works [3].
>
> - **Theorem 4 is, in fact, the most challenging part of this paper.** Due to the page constraints of the main part, this theorem was not emphasized enough. And we believe the techniques used are sufficiently innovative and worth the readers' attention. We briefly introduce the challenges encountered in the algorithm and analysis.
>
>   1. **Algorithmically:** In the strongly convex setting, there is no equivalent adaptive step size or corresponding structure. Therefore, the originally used case-by-case analysis technique [2] can no longer be applied to handle unknown smoothness. To this end, we designed a smoothness estimation method and maintained it using the classic doubling technique in case of inaccurate estimation, as we can calculate the empirical smoothness parameter at the end of each iteration.
>
>   2. **Technically:** In the analysis, we encountered a significant challenge. The inaccurate estimation means cancellations cannot be performed in the analysis, which directly adds a *constant* to the convergence rate, causing the convergence rate in the earlier iterations to seem almost invalidated. Therefore, we turn to consider proving *best-iterate convergence*, and examine when each constant term appears and when it can be optimized to a sufficiently small value. We also employ some relatively novel constructive approaches.
>
>
>
>
> [1] Online to offline conversions, universality and adaptive minibatch sizes, NIPS 2017
>
> [2] UniXGrad: A universal, adaptive algorithm with optimal guarantees for constrained optimization, NeurIPS 2019
>
> [3] Adaptivity and non-stationarity: Problem-dependent dynamic regret for online convex optimization, JMLR 2024
>
> ---
>
> **Q2.** Could the authors elaborate on the specific challenges involved in extending the online-to-batch conversion to the stochastic setting in the strongly convex case? Are there particular obstacles that prevent a direct extension of the techniques used in the smooth or convex cases?
>
> **A2.** Your attention to this gap is very important! As we demonstrated in A1 regarding the challenges and solutions of Theorem 4, the techniques for achieving adaptivity to unknown smoothness in the strongly convex case are, in fact, *completely different* from those in the convex case. As presented in Section 4.2 of our paper, to achieve this result in the strongly convex case, we designed a smoothness observation $L _t$ to guide the updates of our smoothness estimator $\hat{L} _t$. Once we find that the estimator $\hat{L} _t$ is inaccurate, we double it so that it can adaptively approach the underlying smoothness parameter $L _\ell$. The key to this approach is that we have the property $L _t \leq L _\ell$, which ensures that the number of times $\hat{L} _t$ is doubled is limited. However, in the stochastic setting, we can currently only obtain $E[L _t] \leq L _\ell$, which may cause the estimator $\hat{L} _t$ to be excessively large due to noise-induced inaccuracies, ultimately harming our convergence rate. Solving this problem is highly non-trivial and thus beyond the scope of this work and we leave it as an important work for future exploration.
>
> ---
>
> **Q3.** Is it possible to obtain an online-to-batch conversion for Theorem 6 in the case where the curvature parameter (i.e., the strong convexity constant) is unknown? If not, could the authors comment on what makes this technically difficult or whether a universal algorithm could adapt to unknown curvature in this setting?
>
> **A3.** Thanks for the question. To the best of our knowledge, applying ensemble techniques like Theorem 6 to solve offline optimization problems with unknown strong convexity curvature is feasible under Lipschitz conditions. However, under smoothness, since the online-to-batch weights $\alpha _t$ require the knowledge of curvature $\lambda$, which cannot be addressed using universal online learning, solving this problem is challenging and we leave it as a future direction.
>
> ---
>
> If our answers have solved your concerns, we would greatly appreciate it if you could consider a reevaluation of the score for our paper. We are happy to provide further explanations during the upcoming author-reviewer discussions if needed.

---

### Official Review · Reviewer_Agbt · 2025-07-02

**Clarity:** 3
**Significance:** 3
**Originality:** 3
**Rating:** 5
**Confidence:** 3

**Summary:**

The paper develops algorithms for online convex optimization and offline optimization under Hölder smoothness and convexity or strong convexity. Several algorithms are proposed, versions of optimistic gradient descent without and with round weights, and a meta-level learning algorithm. The algorithm does not have to know the class, and it yields optimal bounds in terms of the gradient variation.


Let me go into a bit more detail on how the results are achieved.

Theorems 1 and 3 rely on
(a) the optimistic OMD analysis, resulting in gradient variation appearing plus negative movement terms.
(b) the Hölder-to-quadratic-plus-overhead inequality
(c) decaying learning rate schedule, so that it eventually the negative terms from (a) start helping.

These are all well-known techniques. The contribution of this paper is to show that they can be chained together even in intermediate smoothness regimes i.e. nu in (0,1) and while retaining the gradient variation in the answer.

The results are then applied and extended for offline stochastic optimization, and for dynamic and universal regret.

**Questions:**

Can you relate to the techniques in Wang and Abernethy 2018, Acceleration through Optimistic No-Regret Dynamics?

The presentation of "universality" in online learning is not quite accurate. Going back to van Erven et al (2016), universality is adaptivity to the curvature parameter *without knowing it*. The paper under review instead adapts to a fixed given \lambda or \lambda=0. Wang et al (2019) show how to adapt to the strong convexity parameter (\lambda in the paper under review) without knowing it up-front. Would their approach (of combining multiple learning rates) extend to your setting, and deliver adaptivity to any \lambda without knowing it? That seems like a desirable and within-reach extension.

The universal regret result in Theorem 6 has a *better* exponent on log(T) than the "just-strongly convex" result in Theorem 3. This is strange. What is the catch? Why would we be satisfied with Theorem 3?

**Ethical Concerns:**

["NO or VERY MINOR ethics concerns only"]

**Final Justification:**

I am satisfied with the answers. I hope that the comparison from the rebuttal with the Abernethy et al paper ends up in the paper. The universality claim was my confusion. I hope that the authors can prevent this with a minor adjustment of wording. Finally, I am happy with the authors explanation as to why a better possible dependence on log(T) is possible, and I'm confident they'll implement it.

**Limitations:**

yes

**Paper Formatting Concerns:**

-

**Quality:**

3

**Strengths And Weaknesses:**

Quality:
The achieved adaptivity result is an accomplishment.

Clarity:
The paper is well written and organized clearly. The results are compared to the existing literature.

Significance:
I think this paper will be remembered for its Theorem 6. I find it positively amazing that the Hölder smoothness constants are not required for tuning the algorithm.

Originality:
The paper presents a tighter analysis of an existing algorithm.

---

> ### Author Rebuttal · Authors · 2025-07-30
>
> Thanks for the positive feedback and appreciation of our work! Below we answer your questions and highlight some noteworthy technical contributions of our work.
>
> ---
>
> **Q1.** Can you relate to the techniques in Wang and Abernethy 2018, Acceleration through Optimistic No-Regret Dynamics?
>
> **A1.** Thank you for your reminder! Below is a comparison of the techniques used in the two papers:
>
> **Similarities:** Both Paper [1] and ours utilize an important technique named gradient evaluation on weighted averaged iterates. This technique, combined with smoothness, effectively controls the gradient variation of the optimistic algorithm, and together with negative movement terms, ultimately achieves the accelerated optimization results present in both papers. We will emphasize the connection between our work and theirs [1] regarding this technique in the revised version of the paper.
>
> **Differences:**
>
> - The two papers have different starting points. Our paper starts from online-to-batch conversion [2], applying the adaptivity of online algorithms to offline accelerated optimization; whereas Paper [1] approaches the problem from a game perspective. Both are interesting applications of the optimistic method in online optimization.
> - We not only focus on achieving accelerated optimization but also pay special attention to *adaptively handling unknown smoothness levels*. Theorem 2 presents results adaptive to Hölder smoothness in the convex setting. Theorem 4 provides results adaptive to both smooth and non-smooth cases under the strongly convex setting, which is more challenging. We believe these distinctions are also highly worthy of attention and discussion.
>
>
>
> [1] Wang and Abernethy 2018, Acceleration through Optimistic No-Regret Dynamics
>
> [2] Anytime online-to-batch, optimism and acceleration, ICML 2019
>
> ---
>
> **Q2.** The presentation of "universality" in online learning is not quite accurate. Going back to van Erven et al (2016), universality is adaptivity to the curvature parameter without knowing it. The paper under review instead adapts to a fixed given $\lambda$ or $\lambda=0$. Wang et al (2019) show how to adapt to the strong convexity parameter ($\lambda$ in the paper under review) without knowing it up-front. Would their approach (of combining multiple learning rates) extend to your setting, and deliver adaptivity to any $\lambda$ without knowing it? That seems like a desirable and within-reach extension.
>
> **A2.** Thanks for the question. The problem we are studying here is precisely what was investigated in the previous research on universal online learning. The statement "The paper under review instead adapts to a fixed given $\lambda$ or $\lambda=0$." may stem from some misunderstandings of the paper. The notation of "universal regret" in this paper, i.e., $\text{U-Reg}({\cal F})$, means adapts to unknown function class $\cal F$ which is either convex or strongly convex with **unknown** curvature $\lambda$. Readers might mistakenly interpret ${\cal F}\in\\{{\cal F} _c,{\cal F} _{sc}^\lambda\\}$ that the strong convexity curvature $\lambda$ is known. And as presented in Algorithm 3, page 24, the algorithm only requires domain diameter $D$ and total iterations $T$, without any curvature $\lambda$. Moreover, this algorithm exactly uses the approach of combining multiple learning rates as expected by the reader, and indeed achieves a favorable result.
>
> ---
>
> **Q3.** The universal regret result in Theorem 6 has a better exponent on $\log(T)$ than the "just-strongly convex" result in Theorem 3. This is strange. What is the catch? Why would we be satisfied with Theorem 3?
>
> **A3.** This is a very nice observation! **The conclusion is that we can indeed obtain an $(\log T)^{\frac{1-\nu}{1+\nu}}$ term in Theorem 3.** The reason for the difference — that is, the exponent difference between the $(\log T)^{1-\nu}$ term in current Theorem 3 and the $(\log T)^{\frac{1-\nu}{1+\nu}}$ term in Theorem 6 — is because different *positive and negative cancellation* techniques were used in the proofs of the two theorems, respectively.
>
> - In Theorem 3, we used **Lemma 1**, i.e., $\\|\nabla f({\bf x})-\nabla f({\bf y})\\|^2\lesssim L^2\\|{\bf x-y}\\|^2+L\delta$. And we used the negative movement terms $-\sum _t\\|{\bf x} _t - {\bf x} _{t-1}\\|^2$ to handle the positive terms arising from the decomposition of empirical gradient variation:
>   $$
>   \def \x {\mathbf{x}}
>   \begin{align*}
>   \\|\nabla f _t(\x _t) - \nabla f _{t-1}(\x _{t-1})\\|^2 &\lesssim \\|\nabla f _t(\x _t) - \nabla f _{t-1}(\x _t)\\|^2 + \\|\nabla f _{t-1}(\x _t) - \nabla f _{t-1}(\x _{t-1})\\|^2 \\\\
>   &\lesssim \sup _{\x} \\|\nabla f _t(\x) - \nabla f _{t-1}(\x)\\|^2 + L^2\\|\x _t - \x _{t-1}\\|^2 + L\delta. \tag*{(by Lemma 1)}
>   \end{align*}
>   $$
>
> - In Theorem 6, we used **Lemma 4**, i.e., $\\|\nabla f({\bf x})-\nabla f({\bf y})\\|^2\lesssim L{\cal D} _f({\bf y,x})+L\delta$. And we used the Bregman negativity from linearization $-{\cal D} _{f _t}({\bf x} _\star, {\bf x} _t)$ to handle the positive terms arising from the decomposition of empirical gradient variation:
>   $$
>   \def \x {\mathbf{x}}
>   \def \xs {\x _\star}
>   \begin{align*}
>   \\|\nabla f _t(\x _t) - \nabla f _{t-1}(\x _{t-1})\\|^2 &\lesssim \\|\nabla f _t(\xs) - \nabla f _{t-1}(\xs)\\|^2 + \\|\nabla f _t(\x _t) - \nabla f _t(\xs)\\|^2 + \\|\nabla f _{t-1}(\x _{t-1}) - \nabla f _{t-1}(\xs)\\|^2 \\\\
>   &\lesssim \sup _{\x} \\|\nabla f _t(\x) - \nabla f _{t-1}(\x)\\|^2 + L{\cal D} _{f _t}(\xs, \x _t) + L{\cal D} _{f _{t-1}}(\xs, \x _{t-1}) + L\delta. \tag*{(by Lemma 4)}
>   \end{align*}
>   $$
>
> It can be proven that by using the cancellation from Theorem 6, we can indeed upgrade $(\log T)^{1-\nu}$ to $(\log T)^{\frac{1-\nu}{1+\nu}}$ in Theorem 3 without modifying the algorithm. We will update Theorem 3 to reflect better results in the revised version of the paper. This reflects an interesting insight: we know that, in the smoothness inequality, the form presented in Lemma 1 is indeed looser than that in Lemma 4, and this difference is reflected in the final results of our analysis of Hölder smoothness.
>
> ---
>
> **Additional explanation.** We would also kindly ask you to pay attention to our progress on strongly convex offline optimization, i.e., **Theorem 4**. It is very challenging to develop an algorithm that can adapt to the smoothness level to achieve accelerated convergence while simultaneously guaranteeing nearly optimal convergence in the non-smooth case. To the best of our knowledge, the most recent results date back to the 2017 paper [3], which, however, only obtained *non-accelerated* $O(\exp(-T/\kappa)\cdot T/\kappa)$ result for the smooth case. We improved this to an *accelerated and near-optimal* $O(\exp(-T/(\sqrt{\kappa}\cdot\log \kappa)))$ rate. And we believe the techniques used are sufficiently innovative and worth the readers' attention. We are happy to provide further explanations if needed during the following author-reviewer discussions!
>
> [3] Online to offline conversions, universality and adaptive minibatch sizes, NIPS 2017

---

> > ### Author Response · Authors · 2025-08-06
> >
> > Dear Reviewer Agbt,
> >
> > We sincerely appreciate your recognition of our work and the valuable feedback you have provided. We would also like to ask if you have any comments on our rebuttal, and whether it has addressed the issues you raised. If there are still any concerns, we would be happy to provide further clarification.
> >
> > Best Regards,
> > Authors

---

### Official Review · Reviewer_KXW6 · 2025-07-03

**Clarity:** 2
**Significance:** 3
**Originality:** 3
**Rating:** 5
**Confidence:** 4

**Summary:**

This paper studies online convex optimization under Holder smoothness, and shows that optimistic OGD can obtain $R_T\le O(\sqrt{\sum_t\sup_x\\|f_t(x)-f_{t-1}(x)\\|^2}+T^{\frac{1-\nu}{2}}+\\|\nabla f_1(x_1)\\|)$ regret without any prior knowledge of the Holder smoothness parameters, and it is shown that this yields optimal rates in the stochastic optimization setting. The result is also extended to strongly-convex losses, to time-varying comparators (dynamic regret), and to universal regret.

**Questions:**

- Do you expect that the non-adaptive penalties can be improved to adaptive ones such as $V_T$ or $\sum_t\\|g_t\\|^2$, or are the non-adaptive penalties unavoidable when adapting to the Holder smoothness parameters?

**Ethical Concerns:**

["NO or VERY MINOR ethics concerns only"]

**Final Justification:**

The main weaknesses were that there was essentially one novel result + several "free" ones. The author clarified the difficulties associated with some of the freebies, and theorem 4 seems to be significant. They clarified whether gradient adaptivity *must* be lost, and provided a partial result showing that some adaptivity can be retained, but opens some new questions to be addressed. Overall, well-written paper with a nice technique.

**Limitations:**

yes

**Paper Formatting Concerns:**

No major concerns.

**Quality:**

3

**Strengths And Weaknesses:**

## Strengths

The main contribution in Theorem 1 is very nice and the analysis is very clean. The main tool is Lemma 1, which lets allows you to connect $\\|\nabla f(x)-\nabla f(y)\\|^2$ to $\\|x-y\\|^2$, similar to how you would with smooth losses, but with an additional $\delta$ factor that appears only in the analysis and can therefore be tuned *a posteriori* at the end of the analysis. It's a very nice approach overall and leads to several extensions essentially "for free" after Theorem 1 is established.

## Weaknesses

**Universality**

I think the insistence on defining methods as "strongly universal" will serve only to complicate the literature. For one, there are already other notions of universality, for instance Nesterov's Universal Algorithm, or in OCO algorithms which achieve guarantees on universal regret (as in Section 5.2) are sometimes referred to as being "universal".

But it is also more generally just a short-sighted naming convention. Intuitively, an algorithm should be deemed universal if it solves any problem in a problem class without manually adjusting any problem-dependent parameters; a practitioner should be able to just hit run and know the guarantee will hold. This is not the case here because the results depend on the diameter of the decision set. I think to really be able to call an algorithm universal here you'd have to develop a comparator-adaptive version. So let's suppose that you extend the approach here to achieve comparator adaptivity, what should we call such an algorithm? "Extra Strongly Universal"?

**Loss of Adaptivity**

The key argument seems to unavoidably lead non-adaptive penalties (factors of $T$ instead of, say, $V_T$ or $\sum_t\\|g_t\\|^2$) showing up in the bound. Prior works avoid such penalties from showing up, though they don't adapt to the Holder smoothness parameters; is it possible to improve the results here to achieve fully adaptive bounds? Or does adapting to the Holder smoothness parameters necessitate non-adaptive penalties? For example, the dynamic regret result in the Lipschitz losses setting ends up being $O(G\sqrt{P_T T})$ instead of the usual $O(\sqrt{P_T\sum_t\\|g_t\\|^2})$.

**Minor Weaknesses**

The main novelty is Theorem 1, and each of the extensions are generally very straight-forward afterwards. For example, the extension to stochastic optimization via anytime online-to-batch is well-known from Cutkosky 2019, and the result is basically immediate by combining the online-to-batch argument with Theorem 1. It is also well-known that gradient variation bounds imply improved regret under strong convexity, so these too are fairly easily obtained given Theorem 1, and obtaining the $\sqrt{P_T T}$ dynamic regret bounds is also straight-forward once one has the static regret result.

---

> ### Author Rebuttal · Authors · 2025-07-30
>
> Thanks for the insightful feedback. Below, we will address your questions and clarify our contributions, hoping to address your concerns.
>
> ---
>
> **Q1. Universality.** I think the insistence on defining methods as "strongly universal" will serve only to complicate the literature. For one, there are already other notions of universality, for instance Nesterov's Universal Algorithm, or in OCO algorithms which achieve guarantees on universal regret (as in Section 5.2) are sometimes referred to as being "universal".
>
> **A1.** Thank you for the comment. We indeed only adopt the two existing universal definitions (Nesterov's Universal, and universal in OCO) in our paper. The purpose of introducing Definition 1 (strong and weak universality) in this paper is to facilitate the distinction between the strengths of our methods' adaptability to smoothness. Our Theorem 2 achieves the traditional "Nesterov's Universal," that is, adaptive to Hölder smoothness, while Theorem 4 currently only achieves adaptivity to both smooth and non-smooth cases. Therefore, to help readers differentiate, we refer to Theorem 2 as "strongly universal" and Theorem 4 as "weakly universal." This definition is proposed merely for clarity and is not intended as a contribution of our work.
>
> ---
>
> **Q2. Loss of Adaptivity.** The key argument seems to unavoidably lead non-adaptive penalties (factors of $T$ instead of, say, $V_T$ or $\sum_t\\|g_t\\|^2$) showing up in the bound. Do you expect that the non-adaptive penalties can be improved to adaptive ones such as $V_T$ or $\sum_t\\|g_t\\|^2$, or are the non-adaptive penalties unavoidable when adapting to the Hölder smoothness parameters?
>
> **A2.** Very nice question! Let's take the convex case as an example and see if we can improve our current bound $O(\sqrt{V_T}+T^{\frac{1-\nu}{2}})$ to a more adaptive $O(\sqrt{V_T}+{V_T}^{\frac{1-\nu}{2}})$ or $O(\sqrt{V_T}+(\sum_t\\|g_t\\|^2)^{\frac{1-\nu}{2}})$. First, it is clear that the $O(\sqrt{V_T}+{V_T}^{\frac{1-\nu}{2}})$ bound is impossible, because when $\nu=0$, i.e., in the non-smooth case, this bound becomes $O(\sqrt{V_T})$, which is in fact not achievable. Therefore, we focus on the $O(\sqrt{V_T}+(\sum_t\\|g_t\\|^2)^{\frac{1-\nu}{2}})$ bound. The current conclusion is that we can establish an upper bound that is partially adaptive to $\sum_t\\|g_t\\|^2$, but there are some additional components that are difficult to bound.
>
> We consider still using a method similar to Theorem 1, with the only difference being that at each round $t$, we *no longer use the same global parameters* $(L,\delta)$ when applying Lemma 1. Otherwise, we would unavoidably suffer a $T$ penalty, just like in the original proof.
> $$
> \def \x {\mathbf{x}}
> \begin{align*}
> Reg_T&\lesssim D\sqrt{A_T} - \sum_{t=2}^T\frac{1}{\eta_{t+1}}\\|\x_t - \x_{t-1}\\|^2\tag{Eq. (20)} \\\\
> &\lesssim D\sqrt{A_{t_0}}+D\sqrt{\sum_{t=t_0+1}^T\\|\nabla f_t(\x_t)-\nabla f_{t-1}(\x_t)+\nabla f_{t-1}(\x_t)-\nabla f_{t-1}(\x_{t-1})\\|^2}-\sum_{t=2}^T\frac{1}{\eta_{t+1}}\\|\x_t - \x_{t-1}\\|^2 \\\\
> &\lesssim D\sqrt{A_{t_0}}+D\sqrt{V_T}+D\sqrt{\sum_{t=t_0+1}^T\Big(L_t^2\\|\x_t-\x_{t-1}\\|^2+L_t\delta_t\Big)}-\sum_{t=2}^T\frac{1}{\eta_{t+1}}\\|\x_t - \x_{t-1}\\|^2\tag{Lemma 1} \\\\
> &\lesssim D\sqrt{A_{t_0}}+D\sqrt{V_T}+D^2L_{\max}+\sum_{t=t_0+1}^T\left(L_{\max}-\frac{1}{\eta_{t+1}}\right)\\|\x_t - \x_{t-1}\\|^2+D\sqrt{\sum_{t=t_0+1}^TL_t\delta_t}\tag{$L_{\max}\triangleq\max_t L_t$} \\\\
> &\lesssim D\sqrt{V_T}+D^2L_{\max}+DL_\nu^{\frac{1}{1+\nu}}\sqrt{\sum_{t=t_0+1}^T\delta_t^{\frac{2\nu}{1+\nu}}}.\tag{By a case-by-case analysis}
> \end{align*}
> $$
> In the last inequality above we also apply that for any $\delta_t>0$, $L_t=\delta_t^{\frac{\nu-1}{1+\nu}}L_\nu^{\frac{2}{1+\nu}}$. To ultimately achieve the desired adaptive bound, we select $\delta_t=\delta\\|g_t\\|^{\frac{1+\nu}{\nu}}$ (with $0<\nu\le 1$) and then proceed with our derivation:
> $$
> \def \x {\mathbf{x}}
> \begin{align*}
> Reg_T&\lesssim D\sqrt{V_T}+D^2L_\nu^{\frac{2}{1+\nu}}\delta^{\frac{\nu-1}{1+\nu}}G_{\min}^{\frac{\nu-1}{\nu}}+DL_\nu^{\frac{1}{1+\nu}}\delta^{\frac{\nu}{1+\nu}}\sqrt{\sum_{t=1}^{T}\\|g_t\\|^2}\tag{$G_{\min}\triangleq\min_t\\|g_t\\|$} \\\\
> &\lesssim D\sqrt{V_T}+L_\nu G_{\min}^{\nu-1}D^{1+\nu}\left(\sum_{t=1}^{T}\\|g_t\\|^2\right)^{\frac{1-\nu}{2}}.\tag{Tuning $\delta$}
> \end{align*}
> $$
> Obviously, this result has a drawback: there is a $(\min_t\\|g_t\\|)^{\nu-1}$ term that cannot be bounded. This may be because, in order to obtain an adaptive bound, we use different $(\delta_t, L_t)$ for each round, which contradicts the requirement for a global constant $L$ in the case-by-case analysis technique [1]. We currently do not know how to resolve this issue, and it is also possible that when adapting to the Hölder smoothness parameters, these non-adaptive penalties are unavoidable. Thank you for your very valuable question! We will mention it in the next version of the paper.
>
> [1] UniXGrad: A universal, adaptive algorithm with optimal guarantees for constrained optimization, NeurIPS 2019
>
> ---
>
> **Q3. Minor Weaknesses.** The main novelty is Theorem 1, and each of the extensions are generally very straight-forward afterwards. For example, the extension to stochastic optimization via anytime online-to-batch is well-known from Cutkosky 2019, and the result is basically immediate by combining the online-to-batch argument with Theorem 1. It is also well-known that gradient variation bounds imply improved regret under strong convexity, so these too are fairly easily obtained given Theorem 1, and obtaining the $\sqrt{P_T T}$ dynamic regret bounds is also straight-forward once one has the static regret result.
>
> **A3.** Thanks for the comment. Below, we will explain the contributions of our theorems with some noteworthy points.
>
> 1. **About Theorem 2:** It matches the 2024 state-of-the-art results [2]. A key point here is that: due to the power of our Theorem 1, when combined with the anytime online-to-batch conversion, we can achieve state-of-the-art results in stochastic convex optimization. Our intention is not to claim the result of Theorem 2 as our main contribution, but to illustrate the power of Theorem 1.
> 2. **About Theorem 3:** The method and the corresponding analysis for the strongly convex setting also builds upon our advancements in the convex setup, and the obtained result is new for this problem as well. It demonstrates that the step size structure $\eta_t\propto 1/t$ inherently possesses strong adaptivity.
> 3. **About Theorem 4:** This is the **most challenging** theorem in this paper. Due to the page constraints of the main part, this theorem was not emphasized enough. In terms of results, we significantly improved upon the most recent results [3], which date back to 2017, and achieved an *accelerated and near-optimal* convergence rate. And we believe the techniques used are sufficiently innovative and worth the readers' attention. More explanations will be provided at the end of this rebuttal.
> 4. **About Theorem 5:** It is quite interesting in terms of the result $O(\sqrt{V_T P_T}+P_T^{\frac{1+\nu}{2}}T^{\frac{1-\nu}{2}})$. Although we know that the $O(\sqrt{TP_T})$ bound is worst-case optimal [4], whether the gradient variation bound $O(\sqrt{V_TP_T}+P_T)$ is optimal remains an open problem, especially regarding the term $P_T$ [5]. However, no one had known the interpolation of these two bounds before, and we are the first to achieve it, providing a new perspective on this problem. Technically, we avoid the previous approach of using *fixed* step sizes for the base learners and instead implement a more refined algorithm design with adaptive step sizes to achieve adaptivity to smoothness.
> 5. **About Theorem 6:** It demonstrates that the work [6] actually possesses stronger adaptivity to smoothness.
>
> [2] Universality of adagrad stepsizes for stochastic optimization: Inexact oracle, acceleration and variance reduction, NeurIPS 2024
>
> [3] Online to offline conversions, universality and adaptive minibatch sizes, NIPS 2017
>
> [4] Adaptive online learning in dynamic environments, NeurIPS 2018
>
> [5] Adaptivity and non-stationarity: Problem-dependent dynamic regret for online convex optimization, JMLR 2024
>
> [6] A simple and optimal approach for universal online learning with gradient variations, NeurIPS 2024
>
> ---
>
> **Additional explanation for Theorem 4.** We introduce the challenges encountered in the algorithm and analysis, hoping to provide you with a better understanding of the contributions of this theorem.
>
> - **Algorithmically:** In the strongly convex setting, there is no equivalent adaptive step size or corresponding structure. Therefore, the originally used case-by-case analysis technique [1] can no longer be applied to handle unknown smoothness. To this end, we designed a smoothness estimation method and maintained it using the classic doubling technique in case of inaccurate estimation, as we can calculate the empirical smoothness parameter at the end of each iteration.
>
> - **Technically:** In the analysis, we encountered a significant challenge. The inaccurate estimation means cancellations cannot be performed in the analysis, which directly adds a *constant* to the convergence rate, causing the convergence rate in the earlier iterations to seem almost invalidated. Therefore, we turn to consider proving *best-iterate convergence*, and examine when each constant term appears and when it can be optimized to a sufficiently small value. We also employ some relatively novel constructive approaches.
>
>
> ---
>
> If our answers have solved your concerns, we would greatly appreciate it if you could consider a reevaluation of the score for our paper. We are happy to provide further explanations during the upcoming author-reviewer discussions if needed.

---

> > ### Comment · Reviewer_KXW6 · 2025-08-01
> >
> > Very interesting! Thank you for the analysis of the adaptive case. You've answered my main concerns and I have raised my score

---

> > > ### Author Response · Authors · 2025-08-04
> > >
> > > We appreciate your feedback and acknowledgment of our work! We will mention the analysis of the adaptive case in the paper and continue to reflect on it. Thanks again for your valuable and helpful review.

---

### Official Review · Reviewer_yNEM · 2025-07-03

**Clarity:** 3
**Significance:** 3
**Originality:** 3
**Rating:** 5
**Confidence:** 4

**Summary:**

This paper addresses the challenge of achieving optimal regret bounds in online convex optimization (OCO) under gradient variation measures. Specifically, while prior works in OCO typically assume Lipschitz or first-order smooth conditions, the authors considered  $\nu$-Holder conditions which interpolates these two regimes. The authors propose a parameter-free algorithm based on Online Gradient Descent (OGD) that achieves regret bounds of the order  $O(\sqrt{V_T}+T^{\frac{1-\nu}{2}})$, where $V_T$ is gradient variation defined as the maximum total squared variation of the gradients through iterations. This matches the known lower bounds at the two corner points of Lipschitz and smooth functions. The authors further demonstrate that their method extends to various settings such as offline convex optimization, strongly convex optimization, and dynamic and universal regrets.

**Questions:**

We recommend the authors to address the issues in the weakness section, especially the optimality claim of the proposed results stated in the abstract needs to be appropriately justified. Besides, while the submitted work covers a wide range of extended settings, the explicit formulations in this work contain limitations that were not clearly stated. For instance, in the extension of SCO the authors focused on the settings of first-order oracles, while there has been extensive works on zeroth-order optimization, which has different fundamental sample complexities. The authors also focused on Holder conditions with $\nu\leq 1$, while there are works that explores higher-order smoothness conditions. To name a few, see the references at the end, which are by no means exclusive. We recommend the authors to clarify these limitations, and briefly mention if the proposed results can be extended to those settings. We would be happy to raise the score if the above issues are addressed adequately.

Bach and Perchet. "Highly-smooth zero-th order online optimization"
Akhavan et al. "Exploiting higher order smoothness in derivative-free optimization and continuous bandits"
Novitskii and Gasnikov. "Improved exploiting higher order smoothness in derivative-free optimization and continuous bandit"
Yu et al. "Stochastic Zeroth-Order Optimization under Strongly Convexity and Lipschitz Hessian: Minimax Sample Complexity"
Akhavan and Tsybakov. "Gradient-free stochastic optimization for additive models"

**Ethical Concerns:**

["NO or VERY MINOR ethics concerns only"]

**Final Justification:**

All my earlier comments have been adequately addressed. I appreciate the authors’ careful revisions and have raised my score as previously indicated.

**Limitations:**

yes

**Quality:**

3

**Strengths And Weaknesses:**

Strengths:  The main conceptual contribution, adaptivity to unknown regularity under the gradient variation metric for Holder smooth functions, seem to be novel. The proposed algorithm provides a flexible tradeoff between the required sample complexity and the level of smoothness of the objective function. Rigorous analysis was provided and the achieved sample complexities are presented in clean forms. The paper is generally clear, with good use of tables, figures, and notation to present the main results. Section 2 provides an insightful summary of related works and positions the contribution well.

Weakness:
1. While the achievability results are clearly presented, it is not clear that whether the proposed algorithm achieve optimal regrets besides the two special cases of $\nu=0,1$ as no lower bounds were formally presented as theorems or remarks. If no lower bound is provided, the authors should not claim in the abstract that the proposed algorithm is optimal.
2.  According to the discussion provided by the author, the extended result to SCO seems to be already known in Rodomanov et al 2024.
3. The formulation and assumptions in this work can be better clarified. For instance, while OCO is a well known framework, it is recommended to still provide a brief and concrete description before introducing the formulations for extensions.

---

> ### Author Rebuttal · Authors · 2025-07-30
>
> Thanks for the feedback of our work. We will respond to your questions below.
>
> ---
>
> **Q1.** While the achievability results are clearly presented, it is not clear that whether the proposed algorithm achieve optimal regrets besides the two special cases of $\nu=0,1$ as no lower bounds were formally presented as theorems or remarks. If no lower bound is provided, the authors should not claim in the abstract that the proposed algorithm is optimal.
>
> **A1.** Thank you for your suggestion. Although the regret bound of $O(\sqrt{V_T}+T^{\frac{1-\nu}{2}})$ proposed in Theorem 1 has not been proven optimal, there are two noteworthy points: 1) this bound is optimal in the two special cases $\nu=0$ and $\nu=1$; and 2) the online adaptivity in Theorem 1 can be used to obtain the convergence rate in Theorem 2, which is known to be optimal [1]. We will remove the word "optimal" from Line 8, and focus on investigating the lower bound of the regret in future work. Thank you for your suggestion to improve the rigor of our paper.
>
> [1] Y. Nesterov. "Universal gradient methods for convex optimization problems"
>
> ---
>
> **Q2.** According to the discussion provided by the author, the extended result to SCO seems to be already known in Rodomanov et al 2024 [2].
>
> **A2.** Thanks for the comment. Although two works share some similarities, **we differ from them both in techniques and results.**
>
> **Similarities:** In terms of the algorithm, we both use a *decreasing adaptive step size* and apply *gradient evaluation on weighted averaged iterates*. In analysis, we both adopt the approach of treating Hölder smoothness as inexact smoothness (as shown in Lemma 1), introducing a virtual $(\delta, L)$ to facilitate the final tuning.
>
> **Differences**:
>
> - Techniques: In our analysis, we **explicitly decouple** the two algorithmic components — adaptive step size and gradient evaluation on weighted averaged iterates. We associate the former with achieving smoothness adaptivity in online learning (e.g., Theorem 1), and the latter with the online-to-batch conversion [3], which transforms online learning regret into convergence rates for offline optimization. This makes our proof highly intuitive and has inspired us to **extend this approach to strongly convex optimization**, achieving guarantees (Theorem 4) that significantly improve upon existing results [4]. In contrast, Rodomanov et al. [2] employ a potential-based analysis method without decoupling these two algorithmic components.
>
> - Results: **The bound we obtain for the final tuning $\delta$ differs from theirs**. Specifically, they derive $O(\frac{LD^2}{T^2}+{\color{red}\delta T}+\frac{\sigma D}{\sqrt{T}})$, whereas we obtain $O(\frac{LD^2}{T^2}+{\color{red}\frac{D\sqrt{L\delta}}{\sqrt{T}}}+\frac{\sigma D}{\sqrt{T}})$. This, to some extent, serves as evidence that our analyses differ.
>
> [2] Universality of adagrad stepsizes for stochastic optimization: Inexact oracle, acceleration and variance reduction, NeurIPS 2024
>
> [3] Anytime online-to-batch, optimism and acceleration, ICML 2019
>
> [4] Online to offline conversions, universality and adaptive minibatch sizes, NIPS 2017
>
> ---
>
> **Q3.** The formulation and assumptions in this work can be better clarified. For instance, while OCO is a well known framework, it is recommended to still provide a brief and concrete description before introducing the formulations for extensions.
>
> **A3.** Thank you for your suggestion. In the next version, we will provide readers with a brief and concrete review of the OCO framework before introducing the more complex regret measures and the extension of online-to-batch conversion.
>
> ---
>
> **Q4.** While the submitted work covers a wide range of extended settings, the explicit formulations in this work contain limitations that were not clearly stated. For instance, in the extension of SCO the authors focused on the settings of first-order oracles, while there has been extensive works on zeroth-order optimization, which has different fundamental sample complexities. The authors also focused on Hölder conditions with $\nu\le 1$, while there are works that explores higher-order smoothness conditions. To name a few, see the references at the end, which are by no means exclusive. We recommend the authors to clarify these limitations, and briefly mention if the proposed results can be extended to those settings.
>
> **A4.** Thank you for your suggestion! The direction you proposed, namely zeroth-order SCO under higher-order smoothness, is indeed a potential avenue for further development. We believe that one possible approach is to incorporate a higher-order smoothness form into the gradient variation within the step size design of Eq. (9). This not only retains the adaptivity brought by the adaptive step size decay but also leverages the information from higher-order smoothness, thereby better complementing zeroth-order SCO. In the next version, we will add the following statements:
>
> - In the Introduction section, when presenting the optimization setting (Line 44 and onward), we will emphasize that it is first-order optimization. At the end of the this section, we will add information about zeroth-order optimization and higher-order smoothness:
>   > "In addition to the first-order optimization studied in this paper, the zeroth-order stochastic optimization field is also considering a more general class of smooth settings, namely higher-order smoothness [5,6,7,8,9], which could serve as a potential extension of the methods proposed in this paper."
> - In the conclusion section, we will discuss the potential application of the method proposed in this paper to zeroth-order optimization in the future directions part:
>   > "The adaptivity demonstrated by the method in this paper, with appropriate design, also has the potential to be applied to zeroth-order stochastic optimization under higher-order smoothness conditions."
>   >
>
>
>
> [5] Bach and Perchet. "Highly-smooth zero-th order online optimization"
>
> [6] Akhavan et al. "Exploiting higher order smoothness in derivative-free optimization and continuous bandits"
>
> [7] Novitskii and Gasnikov. "Improved exploiting higher order smoothness in derivative-free optimization and continuous bandit"
>
> [8] Yu et al. "Stochastic Zeroth-Order Optimization under Strongly Convexity and Lipschitz Hessian: Minimax Sample Complexity"
>
> [9] Akhavan and Tsybakov. "Gradient-free stochastic optimization for additive models"
>
> ---
>
> **Additional explanation.** We sincerely appreciate your attention to the convex function setting in our paper (i.e., Theorems 1 and 2). We also recommend paying attention to our progress on strongly convex offline optimization, i.e., **Theorem 4**. It is very challenging to develop an algorithm that can adapt to the smoothness level to achieve accelerated convergence while simultaneously guaranteeing nearly optimal convergence in the non-smooth case. To the best of our knowledge, the most recent results date back to the 2017 paper [4], which, however, only obtained a *non-accelerated* $O(\exp(-T/\kappa)\cdot T/\kappa)$ result for the smooth case. We improved this to an *accelerated and near-optimal* $O(\exp(-T/(\sqrt{\kappa}\cdot\log \kappa)))$ rate. And we are happy to provide further clarifications if needed during the following author-reviewer discussions.
>
> ---
>
> If our answers have solved your concerns, we would greatly appreciate it if you could consider a reevaluation of the score for our paper. We are happy to provide further explanations during the upcoming author-reviewer discussions if needed.

---

### Note · Authors · 2025-08-13

We thank all the reviewers for their valuable suggestions! We summarize our work as follows:
- This paper conducts a systematic study of gradient-variation online learning under Hölder smoothness. Beyond convex functions, we extend the results in three aspects: (i) strongly convex functions, (ii) non-stationary online learning, and (iii) universal online learning.
- By combining these online algorithms with carefully designed online-to-batch conversions, we obtain favorable implications for offline accelerated optimization, including (i) stochastic convex optimization and (ii) deterministic strongly convex optimization.

**Emphasis on one of the main contributions:**
We would like to draw your attention to our progress on strongly convex offline optimization, i.e., **Theorem 4**.
- In terms of results, we developed an algorithm that can adapt to the smoothness level to achieve *accelerated* and *near-optimal* convergence while simultaneously guaranteeing *near-optimal* convergence in the non-smooth case. This improves upon the previously best-known *non-accelerated* result dating back to the 2017 paper [1].
- In terms of techniques, we designed a novel detection-based method. Algorithmically, we dynamically maintained an estimate of the smoothness parameter. Analytically, we provided a more detailed analysis of how estimation errors affect the convergence rate and employed some constructive proof techniques.

Finally, we sincerely thank all the reviewers once again for their valuable suggestions on our work. We will incorporate the corresponding improvements in the next version of the paper. We also appreciate the AC’s time and consideration. We hope this paper will help draw further attention to the direction of bridging online learning and offline optimization.

[1] Online to offline conversions, universality and adaptive minibatch sizes. NIPS 2017

---

### Decision · Program_Chairs · 2025-09-17

**Decision:**

Accept (spotlight)

**Comment:**

This paper investigates online learning with Holder smooth functions. Their main result, Theorem 1, shows that optimistic OGD can obtain a regret $O(D \sqrt{\sum_{x \in X} || \grad f_t(x) - \grad f_{t-1}(x) ||^2 } + L_v D^{1+\nu} T^{(1-\nu)/2} + D || \grad f_1(x_1) ||) on (L_v, v)-holder smooth functions. This result does not require knowledge of v or L_v but does require a domain bound D. This improves over previous online learning bounds for Holder smooth functions.

From this result they are able to achieve several interesting theorems establishing the optimality of optimistic OGD with holder smooth functions in (i) the stochastic setting and (ii) the strongly convex setting.

Minor comment: there appears to be a typo in Theorem 1, in the big-O term L should be L_v